# RETHINKING KL REGULARIZATION IN RLHF: FROM VALUE ESTIMATION TO GRADIENT OPTIMIZATION

## ABSTRACT

Reinforcement Learning from Human Feedback (RLHF) leverages a Kullback-Leibler (KL) divergence loss to stabilize training and prevent overfitting. However, in methods such as GRPO, its implementation may be guided by principles from numerical value estimation—a practice that overlooks the term's functional role as an optimization loss. To analyze this issue, we establish a unified framework that connects two seemingly distinct implementation styles: using the mathematical term $k_n$ as a detached coefficient for the policy's score function ('$k_n$ in reward') or as a direct loss function through which gradients are propagated ('$k_n$ as loss'). We show that the latter can always be analyzed via an equivalent gradient coefficient in the former, unifying the two perspectives. Through this framework, we first prove that conclusions from value estimation fail to guide proper KL loss design, using the '$k_1$ as loss' as a counterexample. We then prove the conventional '$k_1$ in reward' (like PPO) is the principled loss for Reverse KL (RKL) regularization. We further establish a key finding: under on-policy conditions, the '$k_2$ as loss' formulation is, in fact, gradient-equivalent to '$k_1$ in reward'. This equivalence, first proven in our work, identifies both as the theoretically sound implementations of the RKL objective. In contrast, we show that the recently adopted '$k_3$ as loss' (like GRPO) is merely a first-order, biased approximation of the principled loss. Furthermore, we argue that common off-policy implementations of '$k_n$ as loss' methods are biased due to neglected importance sampling, and we propose a principled correction. Our findings provide a comprehensive, gradient-based rationale for choosing and correctly implementing KL regularization, paving the way for more robust and effective RLHF systems.

## 1 INTRODUCTION

The training of state-of-the-art Large Language Models (LLMs) is a multistage process. Following large-scale pretraining and the Supervised Fine-Tuning (SFT) to learn instruction-following behaviors, a final post-training stage, Reinforcement Learning from Human Feedback (RLHF), is often employed. The objective of RLHF is twofold; it serves to align the model more closely with complex human values (Ouyang et al., 2022) and, increasingly, to push the performance limits in specialized reasoning tasks such as mathematics and code generation, as seen in models such as DeepSeek-Math (Shao et al., 2024). A core component of this RLHF process is **KL regularization**, implemented through a loss term derived from the Kullback-Leibler (KL) divergence (Kullback & Leibler, 1951). The **KL loss** serves not only to stabilize the training process but also to improve generalization by preventing the policy from overfitting the reward signal and deviating excessively from the initial SFT model (Ouyang et al., 2022; Stiennon et al., 2020).

Despite the critical role of the KL loss, its theoretical foundations in the optimization context remain underexplored. The choice of its specific mathematical form is often guided by principles from numerical *value estimation*, not from the perspective of gradient-based *optimization*. This category error has led to a proliferation of ad-hoc implementations and suboptimal algorithm designs, exemplified by recent methods like GRPO that adopt sure estimators under the mistaken assumption that good value estimation properties translate to effective gradients. This paper argues that a gradient-centric perspective is essential for designing robust and effective RLHF algorithms.

We perform a systematic, gradient-based analysis of the KL loss to address these issues. We first establish a unified framework that connects two seemingly distinct implementation styles: using the mathematical term $k_n$ as a detached coefficient ('$k_n$ **in reward**') or as a direct loss function ('$k_n$ **as loss**'). This framework allows us to analyze any implementation by examining its equivalent gradient coefficient. Using this lens, we first use the '$k_1$ **as loss**' case as a counterexample to demonstrate the mistakes of the value estimation perspective. We then prove that the conventional '$k_1$ **in reward**' and the '$k_2$ **as loss**' formulations are, in fact, gradient equivalent and represent the principled approach to reverse KL regularization. Finally, we analyze popular alternatives like '$k_3$ **as loss**', revealing their nature as biased approximations, and address a common but critical bug in their off-policy implementation.

Our main contributions are threefold:

1. **A Gradient-Centric Correction for KL Loss Design.** We identify a fundamental flaw in RLHF KL loss design, exemplified by GRPO, where value-estimation principles are misapplied to an optimization objective. We prove this with the '$k_1$ **as loss**' counterexample: despite being an unbiased estimator, its gradient is independent of the reference policy, thus providing no regularization signal.

2. **Identification the Principled KL Loss.** We prove that the conventional '$k_1$ **in reward**' formulation correctly implements the RKL gradient. We further establish a key, previously unrecognized equivalence: '$k_1$ **in reward**' is gradient-equivalent to '$k_2$ **as loss**'. This discovery solidifies both as theoretically sound choices for KL regularization.

3. **Analysis of GRPO Implementations and a Practical Correction.** We analyze popular '$k_3$ **as loss**' used in GRPO is a biased first-order approximation of the principled gradient, leading to weaker regularization or potential instability. Furthermore, we identify a common pitfall in off-policy algorithms where '$k_n$ **as loss**' methods are often implemented without correct importance sampling, and we provide a principled correction for this bias.

## 2 RELATED WORK

**KL Value Estimation.** Since the expectation of the KL divergence is often intractable, it is typically estimated by Monte Carlo sampling. Prior analyses have primarily assessed these estimators as value estimators (John, 2020), characterizing $k_1$ as unbiased but high variance, $k_2$ as biased but lower variance, and $k_3$ as an "optimal", low variance and unbiased choice. We first challenge the claimed superiority of $k_3$ as a value estimator, showing that its advertised properties often fail to hold in practical settings. More importantly, this emphasis is misplaced when these estimators are used for regularization in RLHF. We show that a gradient-centric perspective is essential: conclusions drawn from value estimation do not necessarily translate into effective optimization.

**RLHF Methods.** OpenRLHF (Hu et al., 2024) is the first framework that uses vLLM (Kwon et al., 2023) to accelerate the rollout phase in RLHF training, and incorporates a variety of techniques that make RLHF training more stable. Since then, several training frameworks have emerged, including Verl (Sheng et al., 2024), slime (Zhu et al., 2025), and ROLL (Wang et al., 2025). These frameworks primarily support PPO (Ouyang et al., 2022) and its variants, focusing on improving training stability, particularly addressing challenges in training the critic model. VAPO (Yue et al., 2025) proposed pretraining the critic model to mitigate these issues, while GRPO and Reinforce++ advocate removing it altogether, leading to larger actor model scaling. Most RLHF methods incorporate the KL loss, although some recent rule-based reward algorithms, such as DAPO, have suggested removing the KL loss to enhance performance. However, Prorl (Liu et al., 2025a) solves the performance problem by periodically resetting the reference models, and helps prove where the KL loss still plays a crucial role in preventing overfitting and ensuring long-term training stability. In particular, the KL loss used in Prorl is the '$k_2$ **as loss**' we propose and advocate in this paper.

**KL Loss in RLHF.** The practice of introducing a KL penalty *in the reward* is primarily based on the OpenAI InstructGPT paper (Ouyang et al., 2022), which effectively applies the log-ratio term as a coefficient for the policy's score function, although without a formal justification. Earlier work (Jaques et al., 2019) noted the potential equivalence of adding the KL term to the reward versus the loss, but this was not formally proven. More recently, the GRPO method (Shao et al., 2024),

utilized in influential models such as DeepSeek-R1 (Guo et al., 2025), has gained prominence by adopting the term $k_3$ directly as a KL loss. This choice is justified by citing (John, 2020) and its claim of $k_3$ being an "unbiased estimator", exemplifying the flawed practice of transferring value estimation principles to loss design, a central issue we address.

# 3 PRELIMINARY

## 3.1 VALUE ESTIMATION OF KL DIVERGENCE

The Kullback-Leibler (KL) divergence from a distribution $q(x)$ to a reference $p(x)$ is defined as:

$$D_{\text{KL}}(q \parallel p) = \mathbb{E}_{x \sim q} \left[ \log \frac{q(x)}{p(x)} \right]. \tag{1}$$

As this expectation is often intractable, it is estimated from Monte Carlo samples. Given the importance ratio $\delta(x) = p(x)/q(x)$, common estimators for the term within the expectation include:

$$k_1(x) = -\log \delta(x),$$
$$k_2(x) = \frac{1}{2} \left( \log \delta(x) \right)^2,$$
$$k_3(x) = \delta(x) - 1 - \log \delta(x).$$

Except for the property mentioned in Section 2, the estimator $k_3$ is particularly interesting, designed to reduce the high variance. Although it can be effective when distributions $p$ and $q$ are close, the claim that it is a 'strictly better estimator' (John, 2020) does not hold in the general case. Potential issues of severe bias and infinite variance can arise when the support or tail of the distribution differs significantly. Therefore, its application requires careful verification of some assumptions. A detailed analysis of these statistical instabilities, supported by counterexamples, is provided in Appendix I.

## 3.2 REINFORCEMENT LEARNING FROM HUMAN FEEDBACK (RLHF)

RLHF fine-tunes a policy(actor model) $\pi_\theta$ to produce responses $y$ to prompts $x$ that maximize a reward $r(x, y)$ derived from human preferences. Pure reward maximization may cause reward hacking and distribution drift from a trusted SFT policy. To counteract this, RLHF adds a Reverse KL penalty loss that regularizes the policy toward a fixed reference $\pi_{\text{ref}}$:

$$\mathcal{J}_{\text{RLHF}}(\theta) = \underbrace{\mathbb{E}_{x \sim D, y \sim \pi_\theta(\cdot|x)} \left[ r(x, y) \right]}_{\text{Reward Maximization Term}} - \beta \underbrace{D_{\text{RKL}} \left( \pi_\theta(\cdot|x) \parallel \pi_{\text{ref}}(\cdot|x) \right)}_{\text{KL Regularization Term}} \tag{2}$$
$$= \mathcal{J}_{\text{Reward}}(\theta) - \beta \mathcal{J}_{\text{RKL}}(\theta).$$

Here, $\mathcal{D}$ is the prompt distribution, $y$ is sampled on-policy from the detached snapshot $\pi_\theta$ (numerically equal to $\pi_\theta$ at sampling time), and $\beta$ trades off reward maximization and deviation from $\pi_{\text{ref}}$.

# 4 A UNIFIED FRAMEWORK FOR KL REGULARIZATION IN RLHF

Although most RLHF algorithms optimize for the same high-level objective as Equation (2), their specific implementations differ significantly. To analyze these differences systematically, we establish a unified framework that decomposes the objective into its core components and categorizes the different KL implementation styles.

**Convention** In our analysis, we use $\pi_\theta$ to denote the trainable policy that carries gradients, and follow the standard bandit setting. [1] Samples $y$ are drawn from a detached and numerically identical snapshot policy $\pi_\theta(\cdot|x)$, evaluated in the current iterate; gradients flow only through $\pi_\theta$. All scalar coefficients that multiply the score function $\nabla_\theta \log \pi_\theta(y|x)$ are treated as detached.

---

[1]To simplify analysis of core gradient properties, we model the entire response $y$ as a single action. Our derivations therefore operate on the joint probability $\pi(y|x)$ of the sequence, rather than the token-level probabilities used in standard sequential PPO.

## 4.1 CORE COMPONENTS OF THE RLHF OBJECTIVE

The practical RLHF objective consists of two parts $\mathcal{J}_{\text{Reward}}(\theta)$ and $\mathcal{J}_{\text{KL}}(\theta)$ estimated via Monte Carlo sampling.

**Reward Maximization**  The primary goal is to maximize the expected reward objective, $\mathcal{J}_{\text{Reward}}(\theta) = \mathbb{E}_{x \sim D, y \sim \pi_\theta(\cdot|x)}[r(x,y)]$. As the discrete sampling process $y \sim \pi_\theta$ is non-differentiable, the gradient is estimated via the policy gradient theorem, derived in Appendix B and also known as REINFORCE (Williams, 1992). This approach recasts the gradient as an expectation, enabling its approximation with Monte Carlo samples $y$ drawn from a detached policy snapshot $\pi_\theta$ (numerically equal to $\pi_\theta$ at sampling time). The resulting gradient approximation used for optimization is:

$$\nabla_\theta \mathcal{J}_{\text{Reward}}(\theta) = \mathbb{E}_{x \sim \mathcal{D}, y \sim \pi_\theta(\cdot|x)}[r(x,y) \cdot \nabla_\theta \log \pi_\theta(y|x)]. \tag{3}$$

In practice, $r(x,y)$ represents a shaped advantage signal rather than the raw reward $r_{\text{raw}}(x,y)$ from a reward model, with specifics detailed in Appendix A.

**KL Regularization.**  RLHF regularizes the actor model to the reference model via the RKL:

$$\mathcal{J}_{\text{RKL}}(\theta) = \mathbb{E}_{x \sim \mathcal{D}}[D_{\text{KL}}(\pi_\theta(\cdot|x) \| \pi_{\text{ref}}(\cdot|x))] = \mathbb{E}_{x \sim \mathcal{D}, y \sim \pi_\theta(\cdot|x)}[\log \pi_\theta(y|x) - \log \pi_{\text{ref}}(y|x)]. \tag{4}$$

Analogous to Equation (3), and as derived in Appendix C, RLHF methods utilize the policy gradient trick to formulate a surrogate KL loss. Consequently, expectations are evaluated using Monte Carlo samples $y$ drawn from the detached snapshot $\pi_\theta$ rather than the trainable policy $\pi_\theta$. This surrogate loss incorporates a $k_n$ term; while these terms adopt functional forms from value estimators, they serve a distinct optimization role here and are used primarily to unify the notation. Existing KL loss implementations can be broadly categorized into the following two forms:

1. **$k_n$ as a Detached Coefficient ('$k_n$ in reward'):** Treat $k_n$ as a detached coefficient weight for the score function. A typical choice is $k_1(y|x) = \log \pi_\theta(y|x) - \log \pi_{\text{ref}}(y|x)$ in PPO.

$$\mathcal{J}_{k_n \text{ in reward}}(\theta) := \mathbb{E}_{x \sim \mathcal{D}, y \sim \pi_\theta(\cdot|x)}\left[\underbrace{k_n(\pi_\theta(y|x), \pi_{\text{ref}}(y|x))}_{\text{detached coefficient}} \cdot \log \pi_\theta(y|x)\right]. \tag{5}$$

2. **$k_n$ as a Direct Loss ('$k_n$ as loss'):** Treat $k_n$ as a standalone loss with gradients propagated directly through it. Common choices is $k_3(y|x) = \frac{\pi_{\text{ref}}(y|x)}{\pi_\theta(y|x)} - 1 - \log \frac{\pi_{\text{ref}}(y|x)}{\pi_\theta(y|x)}$ used by GRPO and $k_2(y|x) = \frac{1}{2}\left(\log \frac{\pi_\theta(y|x)}{\pi_{\text{ref}}(y|x)}\right)^2$ we proposed.

$$\mathcal{J}_{k_n \text{ as loss}}(\theta) := \mathbb{E}_{x \sim \mathcal{D}, y \sim \pi_\theta(\cdot|x)}[k_n(\pi_\theta(y|x), \pi_{\text{ref}}(y|x))]. \tag{6}$$

While the gradient of the '$k_n$ as loss' formulation is computed by differentiating $k_n$ directly, under on-policy conditions its gradient can be expressed in the '$k_{n'}$ in reward' form. The equivalent coefficient, $k_{n'}$, is derived as follows:

$$k_{n'}(\pi_\theta(y|x), \pi_{\text{ref}}(y|x)) = \frac{\partial}{\partial \log \pi_\theta} k_n(\pi_\theta(y|x), \pi_{\text{ref}}(y|x)) \tag{7}$$

Crucially, this reveals that $k_{2'}$ is equivalent to $k_1$. We will subsequently prove that under on-policy sampling, '$k_1$ in reward' and '$k_2$ as loss' are gradient-equivalent, principled implementations of the RKL objective in Section 5.2.

## 4.2 KL INTEGRATION FORMS AND ALGORITHM MAPPING

The choice of KL formulation dictates how it is integrated with the reward objective.

**Combined vs. Decoupled Forms.**  Since '$k_n$ in reward' uses the same score function as the reward objective, its coefficient can be merged into the reward coefficient to produce a **Combined Form**—hence the name 'in reward':

$$\mathcal{L}_{\text{Combined}}(\theta) = -\mathbb{E}_{x \sim \mathcal{D}, y \sim \pi_\theta(\cdot|x)}[(r(x,y) - \beta k_n(\pi_\theta(y|x), \pi_{\text{ref}}(y|x))) \cdot \log \pi_\theta(y|x)]. \tag{8}$$

In contrast, '$k_n$ as loss' necessitates a **Decoupled form** with a separate loss form:

$$\mathcal{L}_{\text{Decoupled}}(\theta) = -\mathbb{E}_{x \sim \mathcal{D},\, y \sim \pi_\theta(\cdot|x)} \left[ r(x, y) \cdot \log \pi_\theta(y|x) \right] + \beta\, \mathbb{E}_{x \sim \mathcal{D},\, y \sim \pi_\theta(\cdot|x)} \left[ k_n\big(\pi_\theta(y|x), \pi_{\text{ref}}(y|x)\big) \right]. \quad (9)$$

The decoupled form can also be used with '$k_n$ in reward' by separating the two score function terms. In off-policy updates, the merged coefficient in the combined form, $r - \beta\, k_n$, must be corrected with importance sampling; when combined with PPO, this correction is automatically inherited via the clipped surrogate objective $\pi_\theta / \pi_{\theta_k}$. In contrast, 'as loss' implementations also explicitly require applying IS and PPO clip, but these have always been omitted in practice (see Appendix G).

**Positioning PPO and GRPO.** This framework can position the main algorithms, as summarized in Table 1. PPO is a canonical example of using '$k_1$ in reward' in a combined form. GRPO exemplifies the use of '$k_3$ as loss' in a decoupled form.

Table 1: Decomposition of RLHF algorithms. Note: '$k_n$ in reward' can also be implemented in a decoupled form.

| Algorithm | Typical $k_n$ | KL Formulation Style | Integration Form | Notes on Off-Policy Implementation |
|---|---|---|---|---|
| PPO / REINFORCE | $k_1$ | '$k_n$ in reward' | Combined (typical) | Inherits IS/clipping when paired with PPO. |
| GRPO | $k_3$ | '$k_n$ as loss' | Decoupled | Requires explicit IS/clipping, commonly omitted in practice. |

## 5 GRADIENT-BASED ANALYSIS OF KL IMPLEMENTATIONS

In RLHF, KL regularizers should be selected for gradient properties rather than for accurate estimation of values. In this section, we first use '$k_1$ **as loss**' as a counterexample to show that adopting estimators without auditing the induced gradients can lead to vacuous updates. Then we derive the principal surrogate loss of RKL and demonstrate that '$k_3$ **as loss**' is a first-order approximation. Meanwhile, we also prove that '$k_n$ **as loss**' and '$k_{n'}$ **in reward**' are often gradient equivalent and can be converted to each other with the on-policy setting.

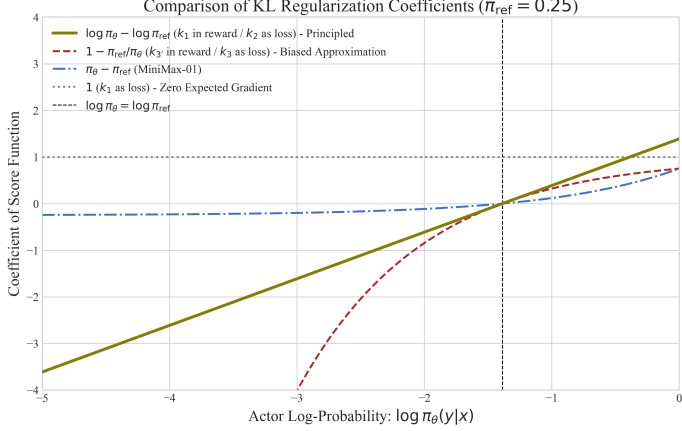

Figure 1: Comparison of KL regularization gradient coefficients. Each curve shows the scalar coefficient $c(x, y)$ which would multiply the score function $\nabla_\theta \log \pi_\theta(y|x)$, plotted against $\log \pi_\theta(y|x)$ with $\pi_{\text{ref}}(y|x) = 0.25$ (vertical dashed line). Principled implementations ('$k_1$ in reward' or '$k_2$ as loss') yield $c = \log\big(\pi_\theta / \pi_{\text{ref}}\big)$, a linear restoring force in log-probability. The '$k_3$ as loss' uses $c = 1 - \pi_{\text{ref}} / \pi_\theta$, a first-order Taylor surrogate of $-\log \delta$ at $\delta = \pi_{\text{ref}} / \pi_\theta = 1$: it is loose when $\log \pi_\theta$ is large ($\pi_\theta \gg \pi_{\text{ref}}$) and can blow up when $\log \pi_\theta$ is small ($\pi_\theta \ll \pi_{\text{ref}}$). The naive '$k_1$ as loss' gives $c \equiv 1$, producing a zero-mean, non-regularizing gradient in expectation.

### 5.1 THE COUNTEREXAMPLE: WHY $k_1$ AS LOSS FAILS

A central lesson of this work is that the desirable properties of *value estimation* do not automatically translate into effective losses of *optimization*. The case of using '$k_1$ **as loss**' provides a clean

counterexample: although $k_1$ is an unbiased estimator of the KL value, it is ineffective as a loss for enforcing a KL constraint.

Consider the direct-loss formulation with on-policy sampling from a detached snapshot $y \sim \pi_\theta(\cdot|x)$:

$$\mathcal{J}_{k_1 \text{ as loss}}(\theta) = \mathbb{E}_{x \sim D, \, y \sim \pi_\theta(\cdot|x)} \big[ \log \pi_\theta(y|x) - \log \pi_{\text{ref}}(y|x) \big]. \tag{10}$$

Since $\pi_{\text{ref}}$ does not depend on $\theta$, its term vanishes upon differentiation, leaving

$$\nabla_\theta \mathcal{J}_{k_1 \text{ as loss}}(\theta) = \mathbb{E}_{x \sim D, \, y \sim \pi_\theta(\cdot|x)} \big[ \nabla_\theta \log \pi_\theta(y|x) \big]. \tag{11}$$

The result exposes a fundamental flaw: the gradient is entirely independent of the reference policy $\pi_{\text{ref}}$; therefore, it carries *no* KL regularization signal.

By the zero-mean score identity in Lemma C.2, the gradient $\nabla_\theta \mathcal{J}_{k_1 \text{ as loss}}(\theta) = 0$, precisely the exact mechanism that underlies the subtraction of the baseline in policy gradients of REIN-FORCE (Williams, 1992). Consequently, in Monte Carlo practice, the term injects only zero-mean noise, which inflates the gradient variance and potentially destabilizes learning.

This counterexample is decisive: an "unbiased value estimator" can produce a useless optimization signal. It directly challenges the assumption that favorable value estimation properties are sufficient for designing effective KL losses, an assumption that has implicitly motivated certain recent implementations, such as GRPO.

### 5.2 The Principled RKL Loss in RLHF: $k_1$ in reward $\Leftrightarrow k_2$ as loss

In the following, we derive the exact on-policy gradient of the Reverse KL objective in Equation (4) and use it as the reference gradient to design surrogates KL loss. Applying the product rule and the log-derivative trick to RKL (see Appendix C) gives:

$$\nabla_\theta \mathcal{J}_{\text{RKL}}(\theta) = \mathbb{E}_{x \sim \mathcal{D}} \left[ \sum_y \nabla_\theta \pi_\theta(y|x) \left( \log \frac{\pi_\theta(y|x)}{\pi_{\text{ref}}(y|x)} + 1 \right) \right]. \tag{12}$$

By the zero-mean score identity in Lemma C.2, the term '+1' vanishes in expectation, resulting in the practical form of the policy gradient:

$$\nabla_\theta \mathcal{J}_{\text{RKL}}(\theta) = \mathbb{E}_{x \sim \mathcal{D}, \, y \sim \pi_\theta(\cdot|x)} \left[ \underbrace{\left( \log \frac{\pi_\theta(y|x)}{\pi_{\text{ref}}(y|x)} \right)}_{k_1 \text{ (detached) coefficient}} \nabla_\theta \log \pi_\theta(y|x) \right]. \tag{13}$$

Any principled KL regularization loss should reproduce this target gradient in expectation. The following theorem shows that two structurally different surrogate losses do so exactly.

**Theorem 5.1** (On-policy gradient equivalence of principled RKL surrogate losses). *Let $\pi_\theta$ be a detached snapshot of the trainable policy $\pi_\theta$ whose parameters coincide at the time of gradient evaluation. For samples $y$ drawn on-policy from $\pi_\theta(\cdot|x)$, the following objectives have the same expected gradient as the target in Equation* (13)*:*

$$\mathcal{J}_{k_1 \text{ in reward}}(\theta) = \mathbb{E}_{x \sim \mathcal{D}, \, y \sim \pi_\theta(\cdot|x)} \left[ \underbrace{\left( \log \frac{\pi_\theta(y|x)}{\pi_{ref}(y|x)} \right)}_{k_1 \text{ (detached) coefficient}} \log \pi_\theta(y|x) \right], \tag{14}$$

$$\mathcal{J}_{k_2 \text{ as loss}}(\theta) = \mathbb{E}_{x \sim \mathcal{D}, \, y \sim \pi_\theta(\cdot|x)} \left[ \frac{1}{2} \left( \log \frac{\pi_\theta(y|x)}{\pi_{ref}(y|x)} \right)^2 \right]. \tag{15}$$

*Sketch (full proof in Appendix C).* For Equation (14), it could recover Equation (13) because the term $k_1$ is a detached scalar multiplying $\nabla_\theta \log \pi_\theta$. For Equation (15), differentiating gives $\nabla_\theta \frac{1}{2} (\log \frac{\pi_\theta}{\pi_{\text{ref}}})^2 = (\log \frac{\pi_\theta}{\pi_{\text{ref}}}) \nabla_\theta \log \pi_\theta$, so here '$k_{2'}$' is '$k_1$', and the general $k_{n'}$ solution formula is in Equation (72), it also yields the same coefficient as Equation (13). $\square$

Consequently, conventional '$k_1$ in reward' (as used in PPO / REINFORCE) and the newly proposed '$k_2$ as loss' are principled, gradient-equivalent, and interchangeable implementations of RKL regularization under on-policy sampling. For off-policy updates, explicit importance sampling and PPO clip are required, as discussed in Appendix G.

## 5.3 First-order approximation of $k_2$ as loss: $k_3$ as loss $\Leftrightarrow k_{3'}$ in reward

$$\mathcal{J}_{k_3 \text{ as loss}}(\theta) = \mathbb{E}_{x \sim \mathcal{D},\, y \sim \pi_\theta(\cdot|x)} \left[ \frac{\pi_{\text{ref}}(y|x)}{\pi_\theta(y|x)} - \log \frac{\pi_{\text{ref}}(y|x)}{\pi_\theta(y|x)} - 1 \right], \qquad (16)$$

where expectations are taken over on-policy samples from the detached snapshot $\pi_\theta$. Taking the gradient of '$k_3$ as loss' yields the equivalent coefficient $k_{3'}$ and could get '$k_{n'}$ in reward' form loss:

$$\nabla_\theta \mathcal{J}_{k_3 \text{ as loss}}(\theta) = \mathbb{E}_{x \sim \mathcal{D},\, y \sim \pi_\theta(\cdot|x)} \Big[ \underbrace{\left(1 - \tfrac{\pi_{\text{ref}}(y|x)}{\pi_\theta(y|x)}\right)}_{k_{3'} \text{ (detached) coefficient}} \nabla_\theta \log \pi_\theta(y|x) \Big] = \nabla_\theta \mathcal{J}_{k_{3'} \text{ in reward}}(\theta).$$

$$(17)$$

Let $\delta = \frac{\pi_{\text{ref}}(y|x)}{\pi_\theta(y|x)}$. The principled '$k_1$ in reward' in Section 5.2 and '$k_{3'}$ in reward' share the same score function but differ only in their scalar coefficients:

$$\textbf{Principled (}k_1\textbf{ in reward / }k_2\textbf{ as loss)} : \quad \underbrace{-\log \delta}_{\text{(detached) coefficient}} \cdot \nabla_\theta \log \pi_\theta(y|x), \qquad (18)$$

$$\textbf{Approximation (}k_3\textbf{ as loss / }k_{3'}\textbf{ in reward)} : \quad \underbrace{-(\delta - 1)}_{\text{(detached) coefficient}} \cdot \nabla_\theta \log \pi_\theta(y|x). \qquad (19)$$

**The Taylor trap.** Around $\delta = 1$, the identity $\log \delta = (\delta - 1) + \mathcal{O}\big((\delta - 1)^2\big)$ implies $-\log \delta \approx 1 - \delta$. Thus, $1 - \delta$ is only a *first-order* surrogate of the principal coefficient $-\log \delta$. The mismatch beyond first-order leads to three concrete issues (see Appendix E for formal statements). For visualization, the coefficient curves are plotted in Figure 1, with the code in Appendix H.

1. **Bias.** For all $\delta \neq 1$, $1 - \delta \neq -\log \delta$, the update direction is biased relative to the true RKL gradient.

2. **Pathological asymmetry.** The two coefficients agree near $\delta = 1$, but behave very differently in the tails:
   - *Over-coverage* $(\delta \to 0)$: $-\log \delta \to +\infty$ (a strong, sustained restoring force), whereas $1 - \delta \to 1$ (saturates), yielding a much weaker regularizer. This often occurs late in RLHF training when $\pi_\theta > \pi_{\text{ref}}$, making the $k_{3'}$ constraint weaker.
   - *Under-coverage* $(\delta \to \infty)$: $-\log \delta$ decays only logarithmically, but $1 - \delta \to -\infty$ much faster, inducing *explosive updates*.

3. **Statistical instability.** Under $y \sim \pi_\theta(\cdot|x)$, $\mathbb{E}[\delta] = 1$ and $\text{Var}[1 - \delta] = \mathbb{E}[(\delta - 1)^2] = \chi^2(\pi_{\text{ref}} \| \pi_\theta)$, the chi-square divergence, which is notoriously unstable. The stochastic gradient inherits this high variance.

## 5.4 Practical Recommendations

Based on our gradient-centric analysis, we offer the following practical recommendations for implementing KL regularization loss in RLHF, and the last two points will be discussed in Appendix G and Appendix F:

**Do not use '$k_1$ as a loss'**. Its expected gradient is zero and independent of the reference model, providing no regularization signal, only noise.

**Prefer '$k_1$ in reward' or '$k_2$ as loss' for theoretical soundness**. In the on-policy setting, these two formulations are gradient equivalent and correctly implement the RKL objective. They are the principal default choices for KL regularization (see Appendix C for a detailed proof).

**Understand the properties of '$k_3$ as loss'**. This formulation should be recognized as a biased first-order approximation of '$k_2$ as loss' (see Appendix Appendix E for a formal analysis). Although its weaker regularization strength at high policy probabilities might offer practical benefits in some scenarios, practitioners should be aware of its theoretical deviation from the true RKL gradient and its potential for pathological updates when the policy probability is low.

**Correct for off-policy bias.** When using any '$k_n$ as loss' formulation in an off-policy setting like PPO, it is crucial to apply importance sampling corrections to the KL term itself. Neglecting this

introduces a systematic bias. The combined approach '$k_n$ in reward' naturally avoids this trap. A detailed discussion and our proposed correction are available in Appendix G.

**Consider bounded alternatives for enhanced stability.** If maximum stability is required, especially under significant policy updates, alternatives that produce bounded gradient coefficients can be beneficial. For example, the MSE-based penalty, like the MiniMax-01 loss, induces a coefficient bounded within $[-1, 1]$. Its derivation and properties are detailed in Appendix F.

# 6 EXPERIMENTAL VALIDATION

We conduct controlled GRPO experiments on a mathematical reasoning task to examine gradient analysis in Section 5. Our experiment design isolates the effects of different KL formulations, allowing us to: (i) validate that '$k_1$ as loss' does not provide a proper regularization term and (ii) compare the principled '$k_2$ as loss' with its first-order surrogate, '$k_3$ as loss'. We put the large-scale experiment in Appendix L and downstream benchmark performance in Appendix M.

## 6.1 SETUP AND BASELINES

**Dataset Construction.** We use a curated subset of OpenR1-Math-220k[2], primarily composed of NuminaMath 1.5 prompts. Reasoning traces are generated by a strong model, Deepseek-R1, and filtered by MathVerify[3] for formatting and correctness. Sequences exceeding 2048 tokens are removed, yielding 7,300 prompts with high-quality off-policy reasoning traces.

**RL Configuration.** To isolate the gradient properties of each KL term, we employ a fully on-policy training configuration, with a rollout batch size of 32, 8 responses per prompt, and an update batch size of 256. The sampling temperature is 1.0. We compute the format reward using regular expressions, use Math-Verify for the accuracy reward, and the actor model is Qwen2.5-Math-1.5B (Yang et al., 2024). We turn off the entropy loss (coefficient 0) and set $\beta = 0.5$ for all KL regularized losses.

## 6.2 KEY RESULTS

The empirical results shown in fig. 2 strongly support our theoretical analysis of '$k_1$ as loss'. As derived in Section 5, the gradient of this loss term is fundamentally flawed for regularization: first, it is entirely independent of the reference policy $\pi_{\text{ref}}$, and second, its expectation over on-policy samples is exactly zero. In practice, this term is equivalent to adding a scaled score function, $\beta \cdot \nabla_\theta \log \pi_\theta$, to the gradient. Although this does not alter the expected update direction, it injects zero-mean noise, thereby increasing gradient variance. It is the inverse of the variance reduction technique used in REINFORCE with a baseline.

Consequently, the theoretical expectation for '$k_1$ as loss' is that its performance will be, at best, comparable to that of having no KL penalty and could potentially be worse due to the increased variance that hinders optimization. Our experimental findings, where the training trajectories of '$k_1$ as loss' are nearly indistinguishable from the baseline without KL, fall squarely within this predicted range of outcomes. This observation provides compelling evidence that '$k_1$ as loss' should be avoided, as it does not benefit regularization while posing a potential risk to the stability of training.

In Figure 3, we compare the principled '$k_2$ as loss' with its approximation, '$k_3$ as loss'. Both methods successfully regularize the policy; a cross-figure comparison with Figure 2 shows that their reward curves are suppressed relative to the baseline without KL, confirming that the KL penalty actively constrains the optimization to stay closer to the reference model.

Both variants regularize the policy, and the principled '$k_2$ as loss' exhibits some advantages. It delivers greater training stability and stronger regularization, reflected in lower variance of rewards and response lengths, indicative of a smoother optimization landscape. It also maintains tighter coupling to the reference policy, with a smaller actor–reference probability gap (see "Logprob Diff with Smooth") and slightly higher entropy, suggesting '$k_2$ as loss' preserves more exploration ability

---

[2]https://huggingface.co/datasets/open-r1/OpenR1-Math-220k
[3]https://github.com/huggingface/Math-Verify

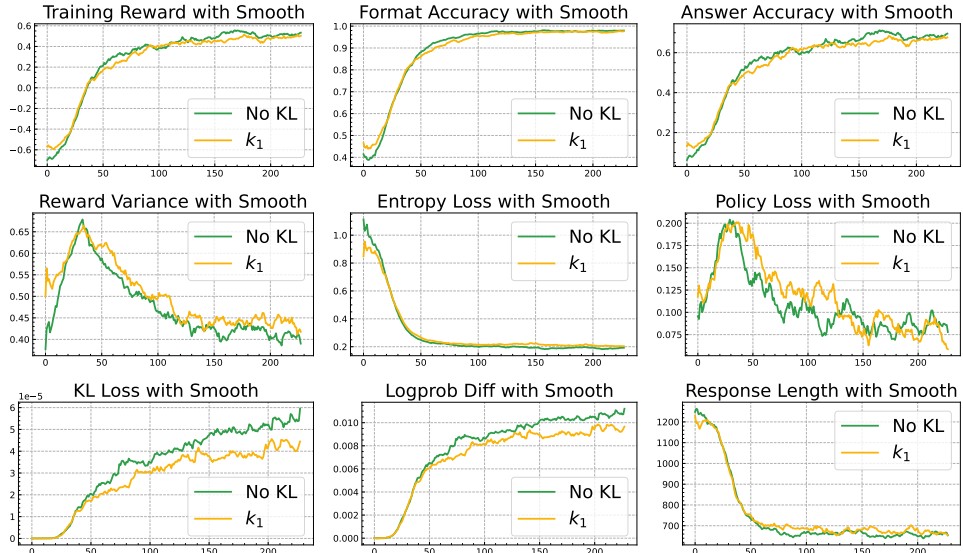

Figure 2: Comparison of '$k_1$ **as loss**' versus no KL regularization. The training dynamics are nearly indistinguishable, empirically confirming the theoretical prediction from Section 5: '$k_1$ **as loss**' is ineffective as a KL regularizer due to its gradient's independence from the reference model and its zero-mean gradient expectation.

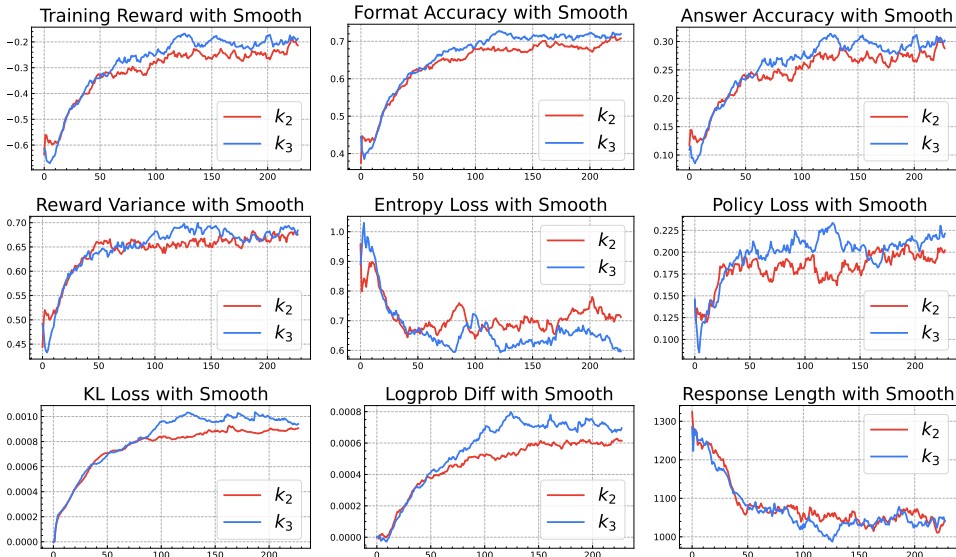

Figure 3: Comparison of the principled '$k_2$ **as loss**' against its first-order surrogate '$k_3$ **as loss**'. Both variants effectively constrain the policy, but '$k_2$ **as loss**' demonstrates superior regularization properties, maintaining a tighter coupling to the reference policy and yielding a more stable optimization path, evidenced by lower reward variance.

while remaining stable. These empirical observations align with its role as the correct surrogate loss for the RKL objective. In contrast, '$k_3$ **as loss**' is a first-order surrogate that imposes weaker constraints, yielding larger probability gaps and reduced entropy. Although '$k_3$ **as loss**' may be a viable choice when a milder late-stage constraint is desired, our results indicate that '$k_2$ **as loss**' offers a more robust and principled route to stable, effective regularization.

## 7 CONCLUSION

We present a systematic, gradient-centric analysis of the KL loss in RLHF, challenging the common GRPO practice of borrowing principles from numerical value estimation to design optimization losses. We established a unified framework that connects the implementations of '$k_n$ **in reward**' and '$k_n$ **as loss**', allowing a direct comparison of their gradient properties. Our analysis identifies conventional '$k_1$ **in reward**' and its newly revealed equivalent, '$k_2$ **as loss**', as the principled loss of Reverse KL loss. And showing the recent '$k_3$ **as loss**' to be a biased first-order approximation of the principal RKL loss. Our experimental results validate these theoretical distinctions. Our work offers a clear and theoretically grounded foundation for implementing KL loss. This addresses long-standing ambiguities in the field and provides practitioners with a robust rationale for designing more effective and reliable RLHF systems.

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

# A    DETAILED IMPLEMENTATION OF RLHF METHODS

In a typical RLHF training step, we draw $N$ prompts $\{x^{(i)}\}_{i=1}^N$ from the dataset $\mathcal{D}$. For each prompt $x^{(i)}$, we sample $G$ responses $y^{(i,1)}, \ldots, y^{(i,G)} \sim \pi_\theta(\cdot \mid x^{(i)})$ from the (detached) current policy. The $G$ responses associated with the same prompt form a *group*, and the full minibatch contains $N \times G$ prompt–response pairs. We use $\pi_\theta$ for the trainable policy (which carries gradients) and $\pi_\theta$ for its numerically identical, detached snapshot (which does not).

**Baseline Subtraction and Normalization**    Proposed by REINFORCE (Williams, 1992), subtracting an action-independent baseline does not change the expected policy gradient (unbiased; see Equation (35)) and typically reduces variance, accelerating convergence . We consider the following baseline operators applied to a scalar signal $r$:

$$f_{\text{group-bl}}(r) = r - \text{mean}_{\text{group}}(r), \tag{20}$$
$$f_{\text{batch-bl}}(r) = r - \text{mean}_{\text{batch}}(r). \tag{21}$$

In practice, normalization is also used. Normalization is *not* unbiased but can improve numerical stability by controlling the scale of the reward signal:

$$f_{\text{BN}}(r) = \frac{r - \text{mean}_{\text{batch}}(r)}{\text{std}_{\text{batch}}(r)}, \tag{22}$$

$$f_{\text{GN}}(r) = \frac{r - \text{mean}_{\text{group}}(r)}{\text{std}_{\text{group}}(r)}. \tag{23}$$

**How Common Algorithms Shape the Reward Signal**    Let $r_{\text{raw}}(x, y)$ denote the raw score from the reward model before any shaping. The following variants differ in how they post-process $r_{\text{raw}}$:

$$\textbf{REINFORCE:} \quad r(x, y) = f_{\text{BN}}\big(r_{\text{raw}}(x, y)\big), \tag{24}$$
$$\textbf{PPO/REINFORCE++:} \quad r(x, y) = f_{\text{BN}}\big(r_{\text{raw}}(x, y)\big), \tag{25}$$
$$\textbf{GRPO:} \quad r(x, y) = f_{\text{GN}}\big(r_{\text{raw}}(x, y)\big), \tag{26}$$
$$\textbf{Dr-GRPO (Liu et al., 2025b):} \quad r(x, y) = f_{\text{group-bl}}\big(r_{\text{raw}}(x, y)\big), \tag{27}$$
$$\textbf{REINFORCE++baseline:} \quad r(x, y) = f_{\text{BN}}\Big(f_{\text{group-bl}}\big(r_{\text{raw}}(x, y)\big)\Big). \tag{28}$$

**Integration of the KL Regularization Loss**    The transforms above shape only the reward signal. The KL regularizer is typically integrated in one of two ways:

(i) **Combined Form ('$k_n$ in reward'):** Used by REINFORCE/PPO methods. A combined reward signal is formed first,

$$A_{\text{combined}}(x, y) = r_{\text{raw}}(x, y) - \beta\, k_1\Big(\pi_\theta(y \mid x),\, \pi_{\text{ref}}(y \mid x)\Big),$$

and then baseline/normalization is applied to this combined signal $A_{\text{combined}}$ before it multiplies the score function.

(ii) **Decoupled Form ('$k_n$ as loss'):** Used by GRPO methods. The KL penalty $k_n\big(\pi_\theta(\cdot \mid x), \pi_{\text{ref}}(\cdot \mid x)\big)$ is optimized as a separate, unnormalized loss term, added to the policy-gradient loss driven by the shaped reward $r(x, y)$.

**Complete On-Policy Objectives for REINFORCE/PPO and GRPO    REINFORCE/PPO (Monte Carlo minibatch):**

$$\mathcal{L}_{\text{REINFORCE/PPO,MC}}(\theta) = -\frac{1}{NG} \sum_{i=1}^N \sum_{j=1}^G \Big\{ A\big(x^{(i)}, y^{(i,j)}\big) \log \pi_\theta\big(y^{(i,j)} \mid x^{(i)}\big) \Big\}, \tag{29}$$

$$\text{where } A(x, y) = f_{\text{BN}}\Big(r_{\text{raw}}(x, y) - \beta\, k_1\Big(\pi_\theta(y \mid x),\, \pi_{\text{ref}}(y \mid x)\Big)\Big). \tag{30}$$

Here, the $k_1$ term is evaluated using detached probabilities, consistent with the policy-gradient framework where it acts as a coefficient for the score function.

**GRPO (Monte Carlo minibatch):**

$$
\mathcal{L}_{\text{GRPO,MC}}(\theta) = -\frac{1}{NG} \sum_{i=1}^{N} \sum_{j=1}^{G} \left\{ r\big(x^{(i)}, y^{(i,j)}\big) \log \pi_\theta\big(y^{(i,j)} \mid x^{(i)}\big) \right\}
$$
$$
+ \frac{\beta}{NG} \sum_{i=1}^{N} \sum_{j=1}^{G} k_3\Big( \pi_\theta\big(y^{(i,j)} \mid x^{(i)}\big),\, \pi_{\text{ref}}\big(y^{(i,j)} \mid x^{(i)}\big) \Big). \tag{31}
$$

In this decoupled form, the shaped reward $r(x,y)$ drives the policy-gradient term, while the KL penalty is a separate loss where gradients flow directly through $\pi_\theta$ inside $k_3(\cdot)$.

*Remark.* While baseline subtraction is an unbiased variance-reduction technique, normalization is a biased but often crucial heuristic for practical stability. Both are important engineering details, even if omitted from simplified theoretical analyses.

## B    Policy Gradient Derivation for Reward Maximization

This section provides a detailed derivation of the policy gradient for the reward maximization objective. We clarify the distinction between the true objective, its gradient, and the surrogate loss function, adhering to a strict notation where $\theta$ indicates a variable subject to differentiation and $\theta$ indicates a detached parameter, such as in a sampling distribution.

**The Objective Function**    The goal is to find parameters $\theta$ for a policy $\pi_\theta$ that maximize the expected reward:

$$
\mathcal{J}_{\text{reward}}(\theta) = \mathbb{E}_{x \sim \mathcal{D}, y \sim \pi_\theta(\cdot|x)} \left[ r(x,y) \right] = \mathbb{E}_{x \sim \mathcal{D}} \sum_y \left[ r(x,y) \cdot \pi_\theta(y|x) \right]. \tag{32}
$$

We assume standard regularity conditions that permit the interchange of differentiation and expectation operators.

**Policy Gradient Derivation**    We compute the gradient of the objective function $\mathcal{J}_{\text{reward}}(\theta)$ using the log-derivative trick. The distinction between $\theta$ and $\theta$ is crucial in the derivation steps:

$$
\begin{aligned}
\nabla_\theta \mathcal{J}_{\text{reward}}(\theta) &= \nabla_\theta \mathbb{E}_{x \sim \mathcal{D}} \sum_y \left[ r(x,y) \cdot \pi_\theta(y|x) \right] \\
&= \mathbb{E}_{x \sim \mathcal{D}} \sum_y \left[ r(x,y) \cdot \nabla_\theta \pi_\theta(y|x) \right] \\
&= \mathbb{E}_{x \sim \mathcal{D}} \sum_y \left[ r(x,y) \cdot \pi_\theta(y|x) \cdot \frac{\nabla_\theta \pi_\theta(y|x)}{\pi_\theta(y|x)} \right] \\
&= \mathbb{E}_{x \sim \mathcal{D}} \sum_y \pi_\theta(y|x) \left[ r(x,y) \cdot \nabla_\theta \log \pi_\theta(y|x) \right] \\
&= \mathbb{E}_{x \sim \mathcal{D}, y \sim \pi_\theta(\cdot|x)} \left[ r(x,y) \cdot \nabla_\theta \log \pi_\theta(y|x) \right].
\end{aligned} \tag{33}
$$

In the third and fourth lines, $\pi_\theta$ represents the current policy's probability value, which is treated as a constant factor in the application of the chain rule for logarithms, while $\pi_\theta$ is the function being differentiated. The final line expresses the gradient as an expectation over samples from $\pi_\theta$, where the sampling process itself is treated as having no gradient path.

## C    Formal Proof of the Principle KL Regularization

This appendix provides a formal, step-by-step derivation showing that, under Assumptions (A1)–(A4), the true KL divergence objective and two common surrogates—'$k_1$ **in reward**' and '$k_2$ **as loss**'—share the same expected gradient. We employ two numerically identical copies of the policy: a gradient-carrying one, $\pi_\theta(\cdot|x)$, and its detached counterpart, $\pi_\theta(\cdot|x)$. Parameters marked with $\theta$ carry gradients; black $\theta$ denotes detached parameters. Sampling measures, denominators, and scalar coefficients multiplying a gradient term are always treated as detached by using $\pi_\theta$.

## C.1 ASSUMPTIONS AND NOTATION

Let $D$ be a data distribution over $x$, $\pi_{\text{ref}}(\cdot|x)$ a fixed reference policy, and $\pi_\theta(\cdot|x)$ a differentiable policy. All logarithms are natural.

**(A1)** For each $x$, the function $y \mapsto \pi_\theta(y|x)$ is a valid probability mass/density: $\pi_\theta(y|x) > 0$ on its support, it is differentiable in $\theta$, and normalizes to one, i.e., $\sum_y \pi_\theta(y|x) = 1$ (or $\int \pi_\theta(y|x)\,dy = 1$).

**(A2)** The interchange of expectation/summation and differentiation is valid.

**(A3)** The data distribution $D$ and reference policy $\pi_{\text{ref}}$ do not depend on $\theta$.

**(A4)** The KL divergence is well-defined: for all $x$ and all $y$ in the support of $\pi_\theta(\cdot|x)$, we have $\pi_{\text{ref}}(y|x) > 0$.

**Notation**: $\overline{\pi_\theta}$ denotes the detached copy of $\pi_\theta$, numerically equal at the current iterate. Expectations over $y$ are taken with respect to $y \sim \pi_\theta(\cdot|x)$ unless stated otherwise.

## C.2 FUNDAMENTAL IDENTITIES

**Lemma C.1** (Log-derivative identity with detached denominator). *For any fixed $x$ and any $y$ with $\pi_\theta(y|x) > 0$,*

$$\nabla_\theta \log \pi_\theta(y|x) = \frac{\nabla_\theta \pi_\theta(y|x)}{\overline{\pi_\theta}(y|x)}. \tag{34}$$

*Proof.* By the chain rule, $\nabla_\theta \log \pi_\theta = (\nabla_\theta \pi_\theta)/\pi_\theta$. Replacing the denominator with its detached, numerically identical copy $\overline{\pi_\theta}$ preserves the numerical value while making the no-gradient path explicit. $\square$ $\square$

**Lemma C.2** (Zero-mean score). *For any fixed $x$,*

$$\mathbb{E}_{y\sim\pi_\theta(\cdot|x)}\big[\nabla_\theta \log \pi_\theta(y|x)\big] = 0. \tag{35}$$

*Proof.* Using Lemma C.1,

$$\sum_y \pi_\theta(y|x)\,\frac{\nabla_\theta \pi_\theta(y|x)}{\overline{\pi_\theta}(y|x)} = \sum_y \nabla_\theta \pi_\theta(y|x) = \nabla_\theta \sum_y \pi_\theta(y|x) = \nabla_\theta(1) = 0. \tag{36}$$

$\square$

**Corollary C.0.1** (Score-function reweighting). *For any function $z(y,x)$ detached with respect to $\theta$,*

$$\sum_y \nabla_\theta \pi_\theta(y|x)\,z(y,x) = \mathbb{E}_{y\sim\pi_\theta(\cdot|x)}\big[z(y,x)\,\nabla_\theta \log \pi_\theta(y|x)\big]. \tag{37}$$

**Corollary C.0.2** (Baseline invariance). *For any function $b(x)$ detached with respect to $\theta$,*

$$\mathbb{E}_{y\sim\pi_\theta(\cdot|x)}\big[b(x)\,\nabla_\theta \log \pi_\theta(y|x)\big] = b(x) \cdot \mathbb{E}_{y\sim\pi_\theta(\cdot|x)}\big[\nabla_\theta \log \pi_\theta(y|x)\big] = 0. \tag{38}$$

*Thus, adding a detached, action-independent baseline $b(x)$ to any coefficient does not change the expected gradient.*

## C.3 DERIVATION OF THE TRUE RKL GRADIENT

The RKL divergence objective is:

$$\mathcal{J}_{\text{RKL}}(\theta) = \mathbb{E}_{x\sim D}\left[\sum_y \pi_\theta(y|x)\,\log\frac{\pi_\theta(y|x)}{\pi_{\text{ref}}(y|x)}\right]. \tag{39}$$

**Step 1 (Differentiate under the expectation).** By (A2), the gradient operator is moved inside the expectation and sum.

**Step 2 (Apply product rule).** For each $y$, we apply the product rule and display detached copies for undifferentiated factors:

$$\nabla_\theta \big[\pi_\theta(y|x) \log \pi_\theta(y|x)\big] = (\nabla_\theta \pi_\theta) \log \pi_\theta + \pi_\theta \cdot \nabla_\theta \log \pi_\theta$$
$$= (\nabla_\theta \pi_\theta)(\log \pi_\theta + 1). \tag{40}$$

By (A3), the gradient of the reference term is $\nabla_\theta[-\pi_\theta(y|x) \log \pi_{\text{ref}}(y|x)] = -(\nabla_\theta \pi_\theta(y|x)) \log \pi_{\text{ref}}(y|x)$.

**Step 3 (Collect terms).** Combining terms yields an expression with a detached coefficient:

$$\nabla_\theta \mathcal{J}_{\text{RKL}}(\theta) = \mathbb{E}_{x \sim D} \left[ \sum_y \nabla_\theta \pi_\theta(y|x) \left( \log \frac{\pi_\theta(y|x)}{\pi_{\text{ref}}(y|x)} + 1 \right) \right]. \tag{41}$$

**Step 4 (Apply score-function reweighting).** Using Corollary C.0.1 with the detached coefficient $z(y,x) := \log \frac{\pi_\theta(y|x)}{\pi_{\text{ref}}(y|x)} + 1$:

$$\nabla_\theta \mathcal{J}_{\text{RKL}}(\theta) = \mathbb{E}_{x \sim D, \, y \sim \pi_\theta(\cdot|x)} \left[ \left( \log \frac{\pi_\theta(y|x)}{\pi_{\text{ref}}(y|x)} + 1 \right) \nabla_\theta \log \pi_\theta(y|x) \right]. \tag{42}$$

**Step 5 (Simplify using the zero-mean score property).** The expectation of the '$+1$' term is zero by Lemma C.2, yielding the final gradient:

$$\nabla_\theta \mathcal{J}_{\text{RKL}}(\theta) = \mathbb{E}_{x \sim D, \, y \sim \pi_\theta(\cdot|x)} \left[ \log \frac{\pi_\theta(y|x)}{\pi_{\text{ref}}(y|x)} \nabla_\theta \log \pi_\theta(y|x) \right]. \tag{43}$$

### C.4 THE GOLD STANDARD: $k_1$ IN REWARD $\Leftrightarrow$ $k_2$ AS LOSS

**Surrogate 1: '$k_1$ in reward'.**

$$\mathcal{J}_{k_1 \text{ in reward}}(\theta) = \mathbb{E}_{x \sim D, \, y \sim \pi_\theta(\cdot|x)} \left[ \underbrace{\left( \log \frac{\pi_\theta(y|x)}{\pi_{\text{ref}}(y|x)} \right)}_{\text{detached coefficient}} \log \pi_\theta(y|x) \right]. \tag{44}$$

Since the coefficient is detached, its gradient is identical to Equation (43).

**Surrogate 2: '$k_2$ as loss'.**

$$\mathcal{J}_{k_2 \text{ as loss}}(\theta) = \mathbb{E}_{x \sim D, \, y \sim \pi_\theta(\cdot|x)} \left[ \frac{1}{2} \left( \log \pi_\theta(y|x) - \log \pi_{\text{ref}}(y|x) \right)^2 \right]. \tag{45}$$

By the chain rule, and displaying the resulting scalar multiplier as its detached copy for clarity:

$$\nabla_\theta \mathcal{J}_{k_2 \text{ as loss}}(\theta) = \mathbb{E}_{x \sim D, \, y \sim \pi_\theta(\cdot|x)} \left[ (\log \pi_\theta(y|x) - \log \pi_{\text{ref}}(y|x)) \nabla_\theta \log \pi_\theta(y|x) \right], \tag{46}$$

which is also identical to Equation (43).

### C.5 CONCLUSION AND IMPLEMENTATION GUIDANCE

Under Assumptions (A1)–(A4), the true KL objective and both surrogates share the same expected gradient, given by Equation (43). This equivalence is based on the following key conventions.

**Sampling Measure**: The samples are drawn from a detached policy $y \sim \pi_\theta(\cdot|x)$(usually vLLM).

**Detached Coefficients**: The scale coefficients that multiply the score function are treated as detached. Applying Corollary C.0.2, any detached baseline $b(x)$ can be added to reduce variance.

**Gradient Path**: Gradients propagate only through terms explicitly parameterized by $\theta$.

**Implementation Notes.** For a single on-policy sample $y \sim \pi_\theta(\cdot|x)$:

- '$k_1$ **in reward**': To minimize $\mathcal{J}_{\text{KL}}$, define the loss as `weight := ` $\log \pi_\theta(y|x) -$ $\log \pi_{\text{ref}}(y|x)$ (detached) and `Loss :=  weight` $\cdot \log \pi_\theta(y|x)$. A gradient descent step on this loss performs descent on $\mathcal{J}_{\text{KL}}$. The common RL loss, $-$`weight` $\cdot \log \pi_\theta$, implements objective *ascent*.
- '$k_2$ **as loss**': To minimize $\mathcal{J}_{\text{KL}}$, define `log_ratio :=` $\log \pi_\theta(y|x) - \log \pi_{\text{ref}}(y|x)$ and `Loss :=` $\frac{1}{2}$ `(log_ratio)`$^2$.

**Remarks.** (i) For continuous spaces, replace sums by integrals; the proof is unchanged provided densities are positive on their support. (ii) The equivalence requires on-policy sampling. If samples are drawn from a stale policy $\pi_{\text{old}}$, exact correction uses importance weights $\rho(x, y) = \pi_\theta(y|x)/\pi_{\text{old}}(y|x)$ inside the expectations.

# D SURROGATE OBJECTIVE: FULL-VOCABULARY VS. MONTE CARLO

This section formalizes the connection between the theoretical policy-gradient objective and its practical mini-batch implementations. We detail two key estimators: a **full-vocabulary loss**, which is exact but computationally infeasible, and a **Monte Carlo (MC) loss**, which provides an unbiased, practical approximation.

**Conventions and gradient paths.**

1. The trainable policy $\pi_\theta$ carries gradients; its detached, numerically identical snapshot at the current iterate is denoted $\pi_\theta$.
2. All scalars that multiply the score function are detached: the reward $r(x, y)$, any KL-derived term $k_n(\cdot)$, and their combination $r(x, y) - \beta\, k_n(\cdot)$.
3. Gradients flow only through $\log \pi_\theta(y|x)$; everything inside the coefficient $c(x, y)$ is detached.
4. We adopt the naming used in the main text: reward (detached) coefficient, $k_n$ (detached) coefficient, and combined form coefficient.

We express the objective using a generic, detached scalar coefficient $c(x, y)$, which can take several forms:

$$c(x, y) \in \left\{ \begin{array}{ll} r(x, y) & \text{reward (detached) coefficient} \\ k_n\big(\pi_\theta(y|x),\, \pi_{\text{ref}}(y|x)\big) & k_n \text{ (detached) coefficient} \\ r(x, y) - \beta\, k_n\big(\pi_\theta(y|x),\, \pi_{\text{ref}}(y|x)\big) & \text{combined form coefficient} \end{array} \right\}. \tag{47}$$

A typical KL choice is

$$k_1(y|x) = \log \pi_\theta(y|x) - \log \pi_{\text{ref}}(y|x). \tag{48}$$

**Policy gradient in expectation form (with baseline).** For the population objective $\mathcal{J}_{\text{true}}(\theta) = \mathbb{E}_{x \sim \mathcal{D},\, y \sim \pi_\theta(\cdot|x)}\big[\, c(x, y)\,\big]$, using a detached, action-independent baseline $b(x)$ and the log-derivative identity with a detached denominator,

$$\nabla_\theta \log \pi_\theta(y|x) = \frac{\nabla_\theta \pi_\theta(y|x)}{\pi_\theta(y|x)}, \tag{49}$$

the unbiased policy gradient is

$$\nabla_\theta \mathcal{J}_{\text{true}}(\theta) = \mathbb{E}_{x \sim \mathcal{D},\, y \sim \pi_\theta(\cdot|x)}\Big[(c(x, y) - b(x))\, \nabla_\theta \log \pi_\theta(y|x)\Big]. \tag{50}$$

This relies on the zero-mean score property under $y \sim \pi_\theta(\cdot|x)$ as proved in Equation (35), ensuring $b(x)$ does not change the expected gradient.

**Population surrogate loss.** A surrogate loss whose negative gradient recovers Equation (50) is

$$\mathcal{L}_{\text{sur}}(\theta) = -\mathbb{E}_{x \sim \mathcal{D},\, y \sim \pi_\theta(\cdot|x)}\Big[\big(c(x, y) - b(x)\big)\, \log \pi_\theta(y|x)\Big]. \tag{51}$$

**Two interchangeable mini-batch implementations.** We provide two equivalent mini-batch estimators of Equation (51). The first computes the exact inner expectation over the discrete action space $\mathcal{V}$ by summing all actions with a detached sampling weight; the second replaces this inner sum with i.i.d. on-policy samples, yielding an unbiased estimate conditional on the mini-batch prompts.

$$\mathcal{L}_{\text{sur,Full}}(\theta) = -\frac{1}{N} \sum_{i=1}^{N} \sum_{y^{(i)} \in \mathcal{V}} \underbrace{\pi_\theta\big(y^{(i)} \mid x^{(i)}\big)}_{\text{detached weight}} \Big(c\big(x^{(i)}, y^{(i)}\big) - b\big(x^{(i)}\big)\Big) \log \pi_\theta\big(y^{(i)} \mid x^{(i)}\big). \quad (52)$$

Here, "detached weight" indicates that gradients do not flow through $\pi_\theta$; the score path is solely via $\log \pi_\theta$.

$$\mathcal{L}_{\text{sur,MC}}(\theta) = -\frac{1}{N} \sum_{i=1}^{N} \frac{1}{G} \sum_{j=1}^{G} \Big(c\big(x^{(i)}, y^{(i,j)}\big) - b\big(x^{(i)}\big)\Big) \log \pi_\theta\big(y^{(i,j)} \mid x^{(i)}\big), \quad y^{(i,j)} \sim \pi_\theta(\cdot|x^{(i)}). \quad (53)$$

In Equation (53), $\{y^{(i,j)}\}_{j=1}^{G}$ are i.i.d. samples from the detached snapshot $\pi_\theta(\cdot|x^{(i)})$; increasing $G$ reduces variance while preserving unbiasedness.

**Unbiasedness and practical considerations.** For any fixed $x^{(i)}$ and function $f$,

$$\mathbb{E}_{\{y^{(i,j)}\}_{j=1}^{G} \text{ i.i.d.} \sim \pi_\theta(\cdot|x^{(i)})} \left[\frac{1}{G} \sum_{j=1}^{G} f\big(y^{(i,j)}\big)\right] = \sum_{y^{(i)} \in \mathcal{V}} \pi_\theta\big(y^{(i)} \mid x^{(i)}\big) f\big(y^{(i)}\big), \quad (54)$$

hence $\mathbb{E}\big[\mathcal{L}_{\text{sur,MC}} \mid \{x^{(i)}\}\big] = \mathcal{L}_{\text{sur,Full}}$. In practice, computing $r(x, y)$ or $k_n(\cdot)$ over the full vocabulary is infeasible for LLMs due to GPU memory constraints; MC estimation is therefore standard.

**Alternative decoupled formulation: '$k_n$ as loss'.** In addition to incorporating KL via the coefficient $c(x, y)$, one may add a separate penalty-only loss that differentiates directly through the log-ratio. Let $\psi_n : \mathbb{R} \to \mathbb{R}$ be differentiable (e.g., $\psi_1(t) = t$, $\psi_2(t) = \frac{1}{2}t^2$). Define

$$\mathcal{L}_{k_n \text{ as loss,MC}}(\theta) = -\frac{1}{NG} \sum_{i=1}^{N} \sum_{j=1}^{G} \psi_n\Big(\underbrace{\log \pi_\theta(y^{(i,j)}|x^{(i)})}_{\text{with grad}} - \underbrace{\log \pi_{\text{ref}}(y^{(i,j)}|x^{(i)})}_{\text{detached}}\Big). \quad (55)$$

Its gradient takes the score-like form

$$\nabla_\theta \mathcal{L}_{k_n \text{ as loss,MC}}(\theta) = -\frac{1}{NG} \sum_{i=1}^{N} \sum_{j=1}^{G} \psi_n'(\cdot) \, \nabla_\theta \log \pi_\theta(y^{(i,j)}|x^{(i)}), \quad (56)$$

where $(\cdot)$ denotes the log-ratio in Equation (55). Here the prime denotes differentiation with respect to the scalar argument:

$$\psi_n'(t) \triangleq \frac{\mathrm{d}}{\mathrm{d}t} \psi_n(t). \quad (57)$$

Under on-policy sampling, evaluating the scalar coefficient $\psi_n'(\cdot)$ at the current iterate (i.e., treating it as detached) yields the gradient-equivalence established in the main text (see Theorem in Section 5.2 and Appendix C). A common choice $\psi_2(t) = \frac{1}{2}t^2$ recovers the squared log-ratio penalty. The total objective is then $\mathcal{L}_{\text{reward,MC}} - \beta \cdot \mathcal{L}_{k_n \text{ as loss,MC}}$, with $\pi_{\text{ref}}$ fixed and detached.

# E  FORMAL ANALYSIS OF THE '$k_3$ AS LOSS' GRADIENT SURROGATE

This appendix provides the formal analysis underpinning Section 5.3, proving that the '$k_3$ as loss' formulation acts as a first-order, biased surrogate for the principled Reverse KL (RKL) gradient. We first derive its gradient-equivalent 'in-reward' coefficient under on-policy sampling, then dissect its three core deficiencies: local bias, global asymmetry, and statistical instability.

Throughout, we fix a prompt $x$ and consider samples $y \sim \pi_\theta(\cdot|x)$ drawn on-policy from a detached snapshot of the trainable policy $\pi_\theta$. We define the probability ratio as:

$$\delta(y) := \frac{\pi_{\text{ref}}(y|x)}{\pi_\theta(y|x)}. \tag{58}$$

Our analysis compares the coefficient induced by '$k_3$ as loss' against the principled RKL gradient coefficient, $c^\star(y) = -\log \delta(y)$. All scalar coefficients multiplying the score function $\nabla_\theta \log \pi_\theta(y|x)$ are treated as detached.

**Gradient-Equivalent Coefficient of '$k_3$ as loss'.** The '$k_3$ as loss' objective is given by:

$$\mathcal{J}_{k_3 \text{ as loss}}(\theta) = \mathbb{E}_{x \sim \mathcal{D}, y \sim \pi_\theta(\cdot|x)} \left[ \frac{\pi_{\text{ref}}(y|x)}{\pi_\theta(y|x)} - 1 - \log \frac{\pi_{\text{ref}}(y|x)}{\pi_\theta(y|x)} \right]. \tag{59}$$

Differentiating and evaluating the resulting scalar multiplier at the detached snapshot yields:

$$\nabla_\theta \mathcal{J}_{k_3 \text{ as loss}}(\theta) = \mathbb{E}_{x \sim \mathcal{D}, y \sim \pi_\theta(\cdot|x)} \left[ \left( 1 - \frac{\pi_{\text{ref}}(y|x)}{\pi_\theta(y|x)} \right) \nabla_\theta \log \pi_\theta(y|x) \right]. \tag{60}$$

This confirms that '$k_3$ as loss' is gradient-equivalent (under on-policy sampling) to an 'in-reward' update with the detached coefficient:

$$c_{3'}(y) := 1 - \delta(y). \tag{61}$$

$$\mathcal{J}_{k_{3'} \text{ in reward}}(\theta) = \mathbb{E}_{x \sim \mathcal{D}, y \sim \pi_\theta(\cdot|x)} \left[ \left( 1 - \frac{\pi_{\text{ref}}(y|x)}{\pi_\theta(y|x)} \right) \log \pi_\theta(y|x) \right]. \tag{62}$$

The remainder of this section formally analyzes the deficiencies of this proxy, $c_{3'}$, when compared to the principled target, $c^\star = -\log \delta$.

**Lemma E.1** (First-order agreement and second-order bias). *The proxy $1 - \delta$ is the first-order Taylor approximation of the principled coefficient $-\log \delta$ around $\delta = 1$. The approximation error (bias) is of second order:*

$$\text{Bias}(\delta) = (-\log \delta) - (1 - \delta) = \frac{1}{2}(\delta - 1)^2 - \frac{1}{3}(\delta - 1)^3 + O((\delta - 1)^4). \tag{63}$$

*Proof sketch.* The result is obtained by expanding $-\log \delta$ in a Taylor series at $r = 1$ and subtracting the term $(1 - \delta)$. $\square$

**Lemma E.2** (One-sided domination and asymmetric tails). *For all $\delta > 0$, the proxy is a strict lower bound, $1 - \delta \leq -\log \delta$, with equality holding only at $\delta = 1$. Their tail behaviors are pathologically asymmetric:*

- ***Over-coverage** ($\delta \to 0^+$): The proxy provides a weak, saturating restoring force ($\lim_{\delta \to 0^+}(1 - \delta) = 1$), whereas the principled coefficient provides an unbounded penalty ($\lim_{\delta \to 0^+}(-\log \delta) = +\infty$).*

- ***Under-coverage** ($\delta \to \infty$): The proxy induces an aggressive, linearly explosive penalty ($\lim_{\delta \to \infty}(1 - \delta) = -\infty$), while the principled coefficient's penalty grows only logarithmically.*

*Proof sketch.* The inequality follows from the fundamental property $\log \delta \leq \delta - 1$. The limits are elementary. $\square$

**Theorem E.1** (Variance equals chi-squared divergence). *Assuming $\text{supp}(\pi_{ref}(\cdot|x)) \subseteq \text{supp}(\pi_\theta(\cdot|x))$, the proxy coefficient $c_{3'}$ has zero mean under the on-policy sampling distribution, and its variance is exactly the chi-squared divergence:*

$$\mathbb{E}_{y \sim \pi_\theta}[1 - \delta(y)] = 0, \qquad \text{Var}_{y \sim \pi_\theta}[1 - \delta(y)] = \chi^2(\pi_{ref}(\cdot|x) \,\|\, \pi_\theta(\cdot|x)). \tag{64}$$

*If the support condition is violated, the variance is infinite.*

*Proof sketch.* $\mathbb{E}[\delta] = \sum_y \pi_\theta(y|x) \frac{\pi_{\text{ref}}(y|x)}{\pi_\theta(y|x)} = 1$, thus $\mathbb{E}[1 - r] = 0$. The variance identity then follows directly from the definition of $\chi^2(p \parallel q)$. $\qquad\square$

**Corollary E.1.1** (Implication for stochastic gradient variance)**.** *The variance of the stochastic gradient term induced by '$k_3$ as loss' is directly governed by the chi-squared divergence, a notoriously unstable metric:*

$$\mathbb{E}\left[\|(1 - \delta(y))\nabla_\theta \log \pi_\theta(y|x)\|^2\right] = \mathbb{E}\left[(1 - \delta(y))^2 \|\nabla_\theta \log \pi_\theta(y|x)\|^2\right]. \tag{65}$$

**Conclusion.** These results provide a rigorous, gradient-centric justification for the claims in the main text. The '$k_3$ as loss' formulation does not implement the true RKL gradient. Instead, it deploys a first-order proxy ($c_{3'} = 1 - \delta$) that is accurate only when the policy is very close to the reference ($\delta \approx 1$). Its pathological tail behavior and high variance, linked to the chi-squared divergence, introduce optimization challenges not present in the principled '$k_1$ in reward' or '$k_2$ as loss' formulations. This analysis underscores the critical importance of selecting regularization losses based on their gradient properties, not merely their characteristics as value estimators.

# F DERIVATION OF AN ALTERNATIVE REGULARIZER: THE MINIMAX-01 LOSS FROM MSE DISTANCE

As an alternative to KL regularization, this section derives the MiniMax-01 loss (Li et al., 2025). We will prove it originates from a mean squared error (MSE) objective and fits within our gradient-centric framework. We adhere to the established on-policy conventions: $\pi_\theta$ is the trainable policy, $\pi_\theta$ is its detached snapshot (numerically equal at the current iterate), and all scalar coefficients that multiply the score function are treated as detached during backpropagation.

**Objective.** We minimize the full-vocabulary MSE between the policy and the reference:

$$\mathcal{J}_{\text{MSE}}(\theta) = \mathbb{E}_{x \sim \mathcal{D}}\left[\frac{1}{2}\sum_y \left(\pi_\theta(y \mid x) - \pi_{\text{ref}}(y \mid x)\right)^2\right]. \tag{66}$$

**On-policy gradient.** Differentiating Equation (66) with respect to $\theta$, evaluating the scalar multiplier at the detached snapshot $\pi_\theta$ (our standard on-policy convention), and converting to the score-function form yields:

$$\begin{aligned}
\nabla_\theta \mathcal{J}_{\text{MSE}}(\theta) &= \mathbb{E}_{x \sim D} \sum_y \left[\frac{1}{2}\nabla_\theta(\pi_\theta(y|x) - \pi_{\text{ref}}(y|x))^2\right] \\
&= \mathbb{E}_{x \sim \mathcal{D}} \sum_y \underbrace{\left(\pi_\theta(y \mid x) - \pi_{\text{ref}}(y \mid x)\right)}_{\text{detached coefficient}} \nabla_\theta \pi_\theta(y \mid x) \\
&= \mathbb{E}_{x \sim \mathcal{D}} \sum_y \pi_\theta(y \mid x) \left(\pi_\theta(y \mid x) - \pi_{\text{ref}}(y \mid x)\right) \nabla_\theta \log \pi_\theta(y \mid x) \\
&= \mathbb{E}_{x \sim \mathcal{D}, y \sim \pi_\theta(\cdot|x)}\left[\left(\pi_\theta(y \mid x) - \pi_{\text{ref}}(y \mid x)\right) \nabla_\theta \log \pi_\theta(y \mid x)\right].
\end{aligned} \tag{67}$$

The last line reveals that MSE regularization induces a score-function update whose scalar coefficient is the probability difference $\pi_\theta - \pi_{\text{ref}}$.

**MiniMax-01 surrogate loss (Monte Carlo).** Using a on-policy sampler that draws $G$ responses $y^{(i,j)} \sim \pi_\theta(\cdot \mid x^{(i)})$ per prompt, the unbiased minibatch surrogate whose negative gradient recovers Equation (67) is

$$\mathcal{L}_{\text{MSE,MC(MiniMax-01)}}(\theta) = -\frac{1}{NG}\sum_{i=1}^N \sum_{j=1}^G \left(\pi_\theta(y^{(i,j)} \mid x^{(i)}) - \pi_{\text{ref}}(y^{(i,j)} \mid x^{(i)})\right) \log \pi_\theta(y^{(i,j)} \mid x^{(i)}).$$

$$\tag{68}$$

This head shares the same in-reward score-function structure as our principled KL implementations: the coefficient is detached, and gradients flow only through $\log \pi_\theta$.

**Key properties and implications.**

1. Bounded gradient coefficient. Since $0 \leq \pi_\theta(y \mid x), \pi_{\text{ref}}(y \mid x) \leq 1$, the coefficient satisfies $-1 \leq \pi_\theta(y \mid x) - \pi_{\text{ref}}(y \mid x) \leq 1$. This boundedness enhances stability against large or pathological updates, in contrast to the unbounded log-ratio used by KL (see Figure 1). This supports our recommendation in Section 5 to consider bounded alternatives when stability is paramount.

2. Symmetry in probability space. The MSE penalty is symmetric with respect to probability differences, providing more conservative corrections when policies diverge, compared to the logarithmic penalty of Reverse KL.

3. Off-policy compatibility. Owing to its in-reward form with a detached coefficient, this head is fully compatible with importance sampling and clipping, following the same correction rules as in Appendix G.

**Remark.** Consistent with our KL analysis, Equation (67) is obtained by evaluating scalar multipliers at the detached snapshot $\pi_\theta$ (on-policy). This keeps all regularizers within a unified $k_n$ multiply score-function lens and enables direct, apples-to-apples comparison of their induced update dynamics.

## G  OFF-POLICY CORRECTION FOR KL REGULARIZATION

Many RLHF implementations harbor a subtle yet critical off-policy bias, particularly when the KL term is implemented "as loss." Such formulations are only gradient-correct under on-policy sampling. For off-policy updates, they require explicit importance sampling (IS) and PPO-style clipping. Omitting these steps systematically biases the update and undermines training stability. This section provides the principled correction, fully aligned with our gradient-centric framework.

### G.1  FROM ON-POLICY TO OFF-POLICY GRADIENTS

We operate in the policy-gradient view, where updates are driven by a detached (stop-gradient) coefficient $c(x, y)$ multiplying the score function. The on-policy gradient estimator is:

$$\nabla_\theta \mathcal{J}_c(\theta) = \mathbb{E}_{x \sim D, \, y \sim \pi_\theta(\cdot \mid x)} \left[ c(x, y) \, \nabla_\theta \log \pi_\theta(y \mid x) \right], \tag{69}$$

where $\pi_\theta$ is a detached snapshot numerically equal to $\pi_\theta$ at the time of gradient evaluation. For samples drawn from a behavior policy $y \sim \pi_{\theta_k}(\cdot \mid x)$, an unbiased off-policy estimator requires IS, assuming the behavior policy has support over the sampled data ($\pi_{\theta_k}(y \mid x) > 0$):

$$\nabla_\theta \mathcal{J}_c(\theta) = \mathbb{E}_{x \sim D, \, y \sim \pi_{\theta_k}(\cdot \mid x)} \left[ \underbrace{\frac{\pi_\theta(y \mid x)}{\pi_{\theta_k}(y \mid x)}}_{\text{detached IS weight}} c(x, y) \, \nabla_\theta \log \pi_\theta(y \mid x) \right]. \tag{70}$$

In practice, PPO replaces this detached IS weight with the gradient-carrying ratio $\rho_k(\theta) = \frac{\pi_\theta(y \mid x)}{\pi_{\theta_k}(y \mid x)}$ (where gradients flow only through the numerator) and employs a clipped surrogate objective to reduce variance. For any detached coefficient $c(x, y)$, the clipped objective to be maximized is:

$$\mathcal{J}_{c, \text{clipped}}(\theta) = \mathbb{E}_{x \sim D, \, y \sim \pi_{\theta_k}(\cdot \mid x)} \left[ \min \left( \rho_k(\theta) \, c(x, y), \, \text{clip}\big(\rho_k(\theta), \, 1 - \epsilon, \, 1 + \epsilon\big) \, c(x, y) \right) \right]. \tag{71}$$

### G.2  CORRECTING "$k_n$ AS LOSS" BY CONVERTING TO AN "IN REWARD" HEAD

A "$k_n$ as loss" head is gradient-correct only on-policy. To adapt it for off-policy use, it must first be converted to its gradient-equivalent "in reward" form. This is achieved by defining a detached (stop-gradient) coefficient $k_{n'}(x, y)$ that reproduces the on-policy gradient of the original loss. For a differentiable penalty $k_n\big(\pi_\theta(y|x), \pi_{\text{ref}}(y|x)\big)$, this coefficient is its derivative with respect to the policy's log-probability, evaluated at the *current* detached snapshot:

$$k_{n'}\big(\pi_\theta(y|x), \pi_{\text{ref}}(y|x)\big) := \left. \frac{\partial}{\partial \log \pi_\theta} k_n\big(\pi_\theta(y|x), \pi_{\text{ref}}(y|x)\big) \right|_{\log \pi = \log \pi_\theta(y|x)}. \tag{72}$$

This conversion precisely aligns with the theoretical equivalences established in the main text:

- **Principled $k_2$ as loss**: $k_2 = \frac{1}{2}(\log \pi_\theta - \log \pi_{\text{ref}})^2 \;\Rightarrow\; k_{2'} = \log \pi_\theta - \log \pi_{\text{ref}}$ (the $k_1$ in reward coefficient).

- **Proxy $k_3$ as loss**: $k_3 = \frac{\pi_{\text{ref}}}{\pi_\theta} - 1 - \log \frac{\pi_{\text{ref}}}{\pi_\theta} \;\Rightarrow\; k_{3'} = 1 - \frac{\pi_{\text{ref}}}{\pi_\theta}$ (the $k_{3'}$ in reward coefficient).

Once expressed as a detached coefficient $k_{n'}(x, y)$, the KL head is handled off-policy exactly like any other score-function head via Equation (71). In PPO with multiple epochs per batch, $k_{n'}$ should be recomputed at each epoch using the updated detached snapshot $\pi_\theta$ to maintain strict gradient equivalence (the denominator $\pi_{\theta_k}$ remains fixed from the rollout).

### G.3 TWO PRINCIPLED OFF-POLICY INTEGRATION STRATEGIES

With the correctly derived coefficient $k_{n'}$ in hand, there are two principled ways to integrate it into the PPO objective, mirroring the on-policy discussion in Section 4.

**1. Combined Form (Single Clipped Head).** Merge the reward advantage and the KL coefficient *before* applying the PPO machinery:

$$A_{\text{combined}}(x, y) \;:=\; r(x, y) \;-\; \beta\, k_{n'}\big(\pi_\theta(y|x), \pi_{\text{ref}}(y|x)\big). \tag{73}$$

The clipped surrogate is then applied to this combined head:

$$\mathcal{J}_{\text{RLHF}}(\theta) = \mathbb{E}_{y \sim \pi_{\theta_k}} \left[ \min\Big( \rho_k(\theta)\, A_{\text{combined}},\; \text{clip}(\rho_k(\theta), 1 - \epsilon, 1 + \epsilon)\, A_{\text{combined}} \Big) \right]. \tag{74}$$

This is the most robust and straightforward approach, as IS and clipping are consistently applied to both components. For correct PPO semantics, form $A_{\text{combined}}$ prior to any baseline subtraction or normalization. This preserves the trade-off set by $\beta$, which would be distorted by shifting or rescaling the KL component.

**2. Decoupled Form (Two Clipped Heads).** Maintain separate reward and KL objectives, each with its own IS correction and clipping scheme:

$$\mathcal{J}_{\text{reward}}(\theta) = \mathbb{E}_{y \sim \pi_{\theta_k}} \left[ \min\Big( \rho_k(\theta)\, r(x, y),\; \text{clip}\big(\rho_k(\theta),\, 1 - \epsilon_1,\, 1 + \epsilon_2\big)\, r(x, y) \Big) \right], \tag{75}$$

$$\mathcal{J}_{\text{KL}}(\theta) = \mathbb{E}_{y \sim \pi_{\theta_k}} \left[ \min\begin{pmatrix} \rho_k(\theta)\, k_{n'}\big(\pi_\theta(y|x), \pi_{\text{ref}}(y|x)\big), \\ \text{clip}\big(\rho_k(\theta),\, 1 - \epsilon,\, 1 + \epsilon\big)\, k_{n'}\big(\pi_\theta(y|x), \pi_{\text{ref}}(y|x)\big) \end{pmatrix} \right]. \tag{76}$$

The final objective to maximize is:

$$\mathcal{J}_{\text{RLHF}}(\theta) \;:=\; \mathcal{J}_{\text{reward}}(\theta) \;-\; \beta\, \mathcal{J}_{\text{KL}}(\theta). \tag{77}$$

This decoupled design affords greater flexibility, such as using asymmetric clipping for the reward head (e.g., $\epsilon_2 > \epsilon_1$) to accelerate learning, while retaining conservative, symmetric clipping for the KL head to ensure stable regularization. Baselines or normalization should be applied only to $A_{\text{reward}}$, not to $k_{n'}(x, y)$, to avoid implicitly altering the regularization strength $\beta$.

**Implementation Notes.**

- **Token vs. Sequence Level:** Our derivations use sequence-level probabilities. In token-level PPO, it is often more stable to compute the ratio as $\rho_k = \exp\big(\sum_t \log \pi_\theta(y_t \mid \cdot) - \sum_t \log \pi_{\theta_k}(y_t \mid \cdot)\big)$ and apply clipping at the sequence level; per-token clipping can be overly conservative.

- **Masking Consistency:** Sum log-probabilities only over action tokens that contribute to the reward/KL (exclude prompt, padding, or masked tokens) to keep $\rho_k$ aligned with the heads being optimized.

- **Numerical Stability and Support:** Ensure $\pi_{\theta_k}(y \mid x) > 0$ for all sampled $(x, y)$ and consider numerically capping $\rho_k$ to prevent overflows under extreme ratios.

- **Adaptive $\beta$:** If targeting a desired KL via an adaptive schedule, update $\beta$ outside the gradient path (detached) and avoid mixing it with advantage normalization; adaptation is orthogonal to IS/clipping and works for both combined and decoupled forms.

**Sign Convention.**    We present objectives for *maximization*. Implementations that *minimize* a loss should negate these expressions, e.g., by minimizing $-\mathcal{J}_{\text{combined}}$ or $-(\mathcal{J}_{\text{reward}} - \beta\,\mathcal{J}_{\text{KL}})$.

## H    VISUALIZATION OF KL REGULARIZATION GRADIENT COEFFICIENTS

To visualize the theoretical arguments discussed in Section 5, this section presents the Python code used to generate Figure 1. The plot compares the behavior of the different scalar coefficients that multiply the policy's score function $\nabla_\theta \log \pi_\theta(y|x)$, as a function of the actor's log-probability for a given token. These coefficients are derived from the respective KL regularization Loss: $k_1$ in reward, $k_2$ as loss, $k_3$ as loss, and the MiniMax-01 loss.

The visualization clearly contrasts the stable, linear behavior of the principled coefficients derived from '$k_1$ **in reward**' / '$k_2$ **as loss**' with the asymmetric behavior of the first-order approximate '$k_3$ **as loss**'. The $k_3$ proxy's tendency to saturate for over-sampled tokens and explode for under-sampled ones, as argued in the main text. The following code, using 'matplotlib' and 'torch', generates the figure.

```python
import torch
import matplotlib.pyplot as plt

# --- Plotting Style ---
plt.style.use('seaborn-v0_8-whitegrid')
plt.rcParams.update({
    "text.usetex": False,  # Disable LaTeX rendering
    "font.family": "serif",  # Use a generic serif font
    "font.serif": ["Times New Roman"],  # Specify Times New Roman as the
        serif font
    "font.size": 14,
    "axes.labelsize": 16,
    "legend.fontsize": 12,
    "xtick.labelsize": 12,
    "ytick.labelsize": 12,
})

# --- Data Generation ---
log_pi_actor = torch.linspace(-5, 0, steps=400)
pi_actor = torch.exp(log_pi_actor)

pi_ref_val = 0.25
log_pi_ref = torch.log(torch.tensor(pi_ref_val))

# --- Coefficients Calculation ---
coeff_k1_loss = torch.ones_like(log_pi_actor)
coeff_k1_reward = log_pi_actor - log_pi_ref
coeff_k3_loss = 1 - pi_ref_val / pi_actor
coeff_minimax = pi_actor - pi_ref_val

# --- Plotting ---
plt.figure(figsize=(10, 6.5))

plt.plot(log_pi_actor, coeff_k1_reward,
         label=r'$\log\pi_{\theta} - \log\pi_{\text{ref}}$ ($k_1$ in
             reward / $k_2$ as loss) - Principled',
         color='#808000', linewidth=3, zorder=10)

plt.plot(log_pi_actor, coeff_k3_loss,
         label=r'$1 - \pi_{\text{ref}}/\pi_{\theta}$ ($k_{3^{\prime}}$ in
             reward / $k_3$ as loss) - Biased Approximation',
         color='Firebrick', linestyle='--', linewidth=2)

plt.plot(log_pi_actor, coeff_minimax,
         label=r'$\pi_{\theta} - \pi_{\text{ref}}$ (MiniMax-01)',
         color='RoyalBlue', linestyle='-.', linewidth=2)
```

```
44
45 plt.plot(log_pi_actor, coeff_k1_loss,
46         label=r'$1$ ($k_1$ as loss) - Zero Expected Gradient',
47         color='Gray', linestyle=':', linewidth=2)
48
49 plt.axvline(x=log_pi_ref.item(), color='black', linestyle='--', linewidth
       =1,
50             label=r'$\log\pi_{\theta} = \log\pi_{\text{ref}}$')
51
52 plt.xlabel(r'Actor Log-Probability: $\log \pi_{\theta}(y|x)$')
53 plt.ylabel(r'Coefficient of Score Function')
54 plt.title(r'Comparison of KL Regularization Coefficients ($\pi_{\text{ref
       }}=0.25$)', fontsize=18)
55 plt.legend(loc='upper left')
56
57 plt.ylim(-4, 4)
58 plt.xlim(-5, 0)
59
60 plt.tight_layout()
61 plt.savefig('comparison_kl_regularization_coefficients.png', dpi=300,
       bbox_inches='tight')
62 plt.show()
```

Listing 1: Python code to generate the comparison plot of KL gradient coefficients.

# I    ON THE STATISTICAL INSTABILITY OF THE $k_3$ VALUE ESTIMATOR

## I.1    THE STRICT PRECONDITION FOR UNBIASEDNESS

An estimator is unbiased if its expectation equals the true value. For $k_3$, the expectation is:

$$\mathbb{E}_q[k_3] = \mathbb{E}_q[\delta(x) - 1 - \log\delta(x)] = (\mathbb{E}_q[\delta(x)] - 1) + D_{KL}(q \parallel p) \tag{78}$$

For $k_3$ to be unbiased, it is necessary that $\mathbb{E}_q[\delta(x)] = 1$. This condition is met if $p$ is absolutely continuous with respect to $q$ ($p \ll q$), which means that the support of $p$ must be contained within the support of $q$.

The condition of a finite KL divergence ($D_{KL}(q \parallel p) < \infty$) is **not sufficient** to guarantee unbiasedness. For example, let $q$ be the uniform distribution in $[0, 1]$ and $p$ be the uniform distribution on $[0, 2]$.

- The KL divergence $D_{KL}(q \parallel p) = \int_0^1 1 \cdot \log(\frac{1}{0.5})dx = \log 2$, which is finite.

- However, $\mathbb{E}_q[r(x)] = \int_0^1 1 \cdot \frac{p(x)}{q(x)}dx = \int_0^1 \frac{0.5}{1}dx = 0.5$.

- The estimator expectation is therefore $\mathbb{E}_q[k_3] = (0.5 - 1) + \log 2 = \log 2 - 0.5$, which is biased.

## I.2    INFINITE VARIANCE AND THE CHI-SQUARED DIVERGENCE

The variance of $k_3$ is dominated by the second moment of the importance ratio, $\mathbb{E}_q[\delta(x)^2]$. This term is directly related to the Chi-squared divergence.

When $p \ll q$, the identity holds: $\chi^2(p \parallel q) = \mathbb{E}_q[(\delta(x)-1)^2] = \mathbb{E}_q[\delta(x)^2]-1$. If $p$ is not absolutely continuous with respect to $q$ ($p \not\ll q$), $\chi^2(p \parallel q)$ is defined to be infinite.

Therefore, the variance of $k_3$ will be infinite if $\mathbb{E}_q[\delta(x)^2]$ is infinite. This occurs if $p \not\ll q$ or if $p \ll q$ but the tails of $q$ are sufficiently lighter than the tails of $p$. While the divergence of $\mathbb{E}_q[\delta(x)^2]$ is the primary cause of instability, the finiteness of $\text{Var}(k_3)$ also technically requires the finiteness of $\mathbb{E}_q[(\log\delta(x))^2]$.

## I.3 THE GAUSSIAN CASE AND AN EMPIRICAL DEMONSTRATION

For two Gaussian distributions, $p \sim \mathcal{N}(\mu_p, \sigma_p^2)$ and $q \sim \mathcal{N}(\mu_q, \sigma_q^2)$, the variance of $k_3$ is finite if and only if $\sigma_q^2 > \sigma_p^2/2$. This condition illustrates that the sampling distribution $q$ must be sufficiently "wide" relative to the reference distribution $p$. This condition generalizes for multivariate Gaussians with covariance matrices $\Sigma_p$ and $\Sigma_q$. **The expectation $\mathbb{E}_q[r(x)^2]$ is calculated via an integral involving the ratio of two Gaussian probability densities. For this integral to converge (and thus for the variance to be finite), it is required that the matrix $2\Sigma_q - \Sigma_p$ be positive definite.**

This failure mode is empirically illustrated below, where a narrow Gaussian $q(x)$ ($\sigma_q = 0.2$) is used to estimate the KL divergence to a standard Gaussian $p(x)$ ($\sigma_p = 1$). This configuration violates the condition, since $0.2^2 \not> 1^2/2$.

```python
import torch
import torch.distributions as dist

# p: reference distribution, q: sampling distribution
p = dist.Normal(loc=0, scale=1)
q = dist.Normal(loc=0.1, scale=0.2) # A narrow distribution where Var[k3]
    is infinite

# Sample from the narrow distribution q
x = q.sample(sample_shape=(10_000,))

# Ground truth KL divergence D_KL(q || p)
true_kl = dist.kl_divergence(q, p)

# Compute the log-ratio log(p(x)/q(x))
log_r = p.log_prob(x) - q.log_prob(x)
r = torch.exp(log_r)

# Define estimators
k1 = -log_r
k2 = log_r.pow(2) / 2
k3 = r - 1 - log_r

# --- Code to generate output ---
print(f"True KL Divergence: {true_kl:.4f}\n")
print("Estimator         | Sample Mean    | Sample Std. Dev.")
print("------------------|----------------|------------------")
estimators = {"k1": k1, "k2": k2, "k3": k3}

for name, k in estimators.items():
    mean = k.mean()
    std = k.std()
    print(f"{name:<17} | {mean:>13.4f} | {std:>16.4f}")

# --- Actual Output 1 ---
True KL Divergence: 1.1344

Estimator         | Sample Mean    | Sample Std. Dev.
------------------|----------------|------------------
k1                |         1.1272 |           0.6912
k2                |         0.8742 |           0.6006
k3                |         0.8136 |           8.8244
# --- Actual Output 2 ---
True KL Divergence: 1.1344

Estimator         | Sample Mean    | Sample Std. Dev.
------------------|----------------|------------------
k1                |         1.1336 |           0.6611
k2                |         0.8611 |           0.5210
k3                |         0.6817 |           4.1082
# --- Actual Output 3 ---
```

```
51  True KL Divergence: 1.1344
52
53  Estimator           | Sample Mean   | Sample Std. Dev.
54  ------------------|---------------|------------------
55  k1                  |        1.1348 |           0.6709
56  k2                  |        0.8689 |           0.4968
57  k3                  |        0.6595 |           1.4925
58  # --- Actual Output 4 ---
59  True KL Divergence: 1.1344
60
61  Estimator           | Sample Mean   | Sample Std. Dev.
62  ------------------|---------------|------------------
63  k1                  |        1.1256 |           0.6962
64  k2                  |        0.8758 |           0.6263
65  k3                  |        0.9772 |          26.7379
```

Listing 2: Code illustrating the high variance of the $k_3$ value estimator when the sampling distribution q(x) is too narrow.

The results vividly illustrate the issue. The sample standard deviation of $k_3$ is **several times larger** than that of $k_1$. And several repeated experiments show that the numerical instability of $k_3$ is obviously more severe than $k_1$ and $k_2$. The large gap between the sample mean of $k_3$ and the true KL value is not estimator bias, but rather a large **sampling error**, which is characteristic of an estimator with immense or infinite variance. This demonstrates that an impractically large number of samples would be required for the estimate to converge reliably, making $k_3$ an unreliable choice in such scenarios.

## J GROUP NORMALIZATION STABILITY ISSUES

GRPO performs per-prompt group normalization: for a prompt with $G$ responses and rewards $\mathbf{r} = \{r_1, \ldots, r_G\}$, the advantage is

$$A_i = \frac{r_i - \text{mean}_{\text{group}}(\mathbf{r})}{\text{std}_{\text{group}}(\mathbf{r})}. \tag{79}$$

**Stability issue.** When the within-group variance is very small (e.g., $\mathbf{r} = [0.99999, 1.00001, 0.99999, 1.00001]$), normalization can dramatically amplify tiny numerical differences. For the above example, the resulting advantages become approximately $[-0.8660, 0.8660, -0.8660, 0.8660]$ (using the unbiased sample standard deviation), which destabilizes optimization by turning near-constant rewards into large-magnitude updates.

**Proposed solution.** Clip the standard deviation to prevent pathological amplification:

$$A_i = \frac{r_i - \text{mean}_{\text{group}}(\mathbf{r})}{\text{clip\_std}_{\text{group}}(\mathbf{r})}, \qquad \text{clip\_std}_{\text{group}}(\mathbf{r}) = \max\big(\min(\text{std}_{\text{group}}(\mathbf{r}), \text{std}_{\max}), \text{std}_{\min}\big). \tag{80}$$

Here, $\text{std}_{\min} > 0$ is a small floor that prevents exploding normalization when variance collapses, and $\text{std}_{\max}$ avoids under-normalization when variance is unusually large. In practice, setting $\text{std}_{\min}$ as a small constant relative to the reward scale (e.g., $10^{-1}$) may be effective.

**Why this matters beyond binary rewards.** Although binary 0/1 rewards in RLVR can sometimes mitigate extreme cases, more general regression reward models—such as those trained with Bradley–Terry (BT) losses—often produce continuous scores that may become highly concentrated (e.g., near 0 or 1) on easy or very hard prompts. In such regimes, within-group standard deviations can be arbitrarily small even when rewards are bounded in $[0, 1]$, and group normalization will over-scale negligible differences unless a variance floor (or clipping) is used. Therefore, std clipping is important not only for numerical stability but also to avoid over-amplifying noise when reward predictions saturate.

**Remark.** For reward scores bounded in $[0, 1]$, $\mathrm{std}(\mathbf{r}) < 1$ always holds, but it can be orders of magnitude smaller than 1 in practice; the smaller the variance, the stronger the amplification effect from group normalization. Clipping $\mathrm{std}_{\mathrm{group}}(\mathbf{r})$ preserves the intended scale-invariance when variance is moderate, while guarding against instability when variance collapses.

# K FORWARD KL VS. REVERSE KL: A GRADIENT-CENTRIC REINTERPRETATION

While our analysis primarily centers on the Reverse KL (RKL) divergence—the standard regularization objective in RLHF—recent perspectives, such as those by Tang & Munos (2025), point out that the GRPO-style '$k_3$ as loss' formulation actually optimizes the KL divergence with swapped arguments, $D_{\mathrm{KL}}(\pi_{\mathrm{ref}}\|\pi)$. While Tang & Munos (2025) refer to this as "reversed KL" relative to the RKL target, in the broader variational inference literature, this objective is known as the **Forward KL (FKL)**.

In this section, we refine this insight using our unified gradient framework. We demonstrate that '$k_3$ as loss' is not merely an estimator for the FKL gradient, but specifically a **variance-reduced estimator with an implicit baseline**. This perspective clarifies its local variance-reduction properties near the reference policy despite the objective mismatch, while highlighting the geometric risks (mean-seeking behavior) in the tails.

**Derivation of the FKL Gradient.** The Forward KL objective is defined as:

$$\mathcal{J}_{\mathrm{FKL}}(\theta) = \mathbb{E}_{x\sim\mathcal{D}}\Big[D_{\mathrm{KL}}\big(\pi_{\mathrm{ref}}(\cdot|x)\,\|\,\pi_\theta(\cdot|x)\big)\Big] = \mathbb{E}_{x\sim\mathcal{D},\,y\sim\pi_{\mathrm{ref}}(\cdot|x)}\big[\log\pi_{\mathrm{ref}}(y|x) - \log\pi_\theta(y|x)\big]. \tag{81}$$

Rewriting the gradient expectation using importance sampling over the current policy $\pi_\theta$ (to allow on-policy estimation) yields the standard FKL policy gradient:

$$\nabla_\theta\mathcal{J}_{\mathrm{FKL}}(\theta) = \mathbb{E}_{x\sim\mathcal{D},\,y\sim\pi_\theta(\cdot|x)}\Big[\underbrace{-\frac{\pi_{\mathrm{ref}}(y|x)}{\pi_\theta(y|x)}}_{\text{Standard FKL coeff. } (-\delta)}\nabla_\theta\log\pi_\theta(y|x)\Big], \qquad \text{where } \delta := \frac{\pi_{\mathrm{ref}}(y|x)}{\pi_\theta(y|x)}. \tag{82}$$

**$k_3$ as Loss: FKL with an Implicit Baseline.** Recalling Equation (17), the gradient induced by the '$k_3$ as loss' formulation uses the coefficient $1 - \delta$. Comparing this with the standard FKL coefficient in Equation (82) reveals an implicit decomposition:

$$\nabla_\theta\mathcal{J}_{k_3\text{ as loss}}(\theta) = \mathbb{E}_{x\sim\mathcal{D},\,y\sim\pi_\theta(\cdot|x)}\Big[\big(\underbrace{(-\delta)}_{\text{Standard FKL}} - \underbrace{(-1)}_{\text{Implicit Baseline } b}\big)\nabla_\theta\log\pi_\theta(y|x)\Big]. \tag{83}$$

Although mathematically equivalent in expectation (since $\mathbb{E}[\nabla\log\pi] = 0$), the implicit baseline $b = -1$ acts as a **control variate**:

- **Variance Reduction at Convergence:** Near the reference policy ($\pi_\theta \approx \pi_{\mathrm{ref}}$), we have $\delta \approx 1$. The standard FKL estimator $-\delta$ fluctuates around $-1$ (high variance), whereas the $k_3$ coefficient $(1 - \delta)$ fluctuates around 0. This explains why $k_3$ **exhibits low variance specifically** in the low-KL regime: it is effectively a zero-variance estimator of the FKL gradient at the optimum.

**Geometric Implications: Mean-Seeking vs. Mode-Seeking.** While the implicit baseline stabilizes estimation, optimizing FKL fundamentally alters the regularization geometry compared to the principled RKL:

- **RKL (Mode-seeking):** As derived in Section 5.2, RKL uses $-\log\delta$. As $\delta \to 0$ (policy places mass where reference does not), $-\log\delta \to \infty$. This creates a "barrier" that forces the policy to stay within the reference's support.
- **FKL (Mean-seeking):** The $k_3$ coefficient $1 - \delta$ saturates at 1 as $\delta \to 0$. This "mean-seeking" behavior imposes only a finite penalty for generating out-of-distribution tokens.

Consequently, under distribution shift, '$k_3$ as loss' fails to strongly penalize drift into regions unsupported by the reference model, confirming our experimental findings of higher variance and weaker constraint compliance.

In summary, '$k_3$ as loss' can be seen as a statistically coherent, baseline-corrected estimator of the FKL gradient, with favorable variance properties near the reference policy. However, its underlying geometry (mean-seeking and mode-covering) remains fundamentally different from that of RKL (mode-seeking). For RLHF applications where keeping the policy tightly constrained within the support of the reference model is a primary concern, the principled RKL implementations discussed in Section 5.2 may offer stronger and more reliable regularization.

## L LARGE SCALE EXPERIMENT

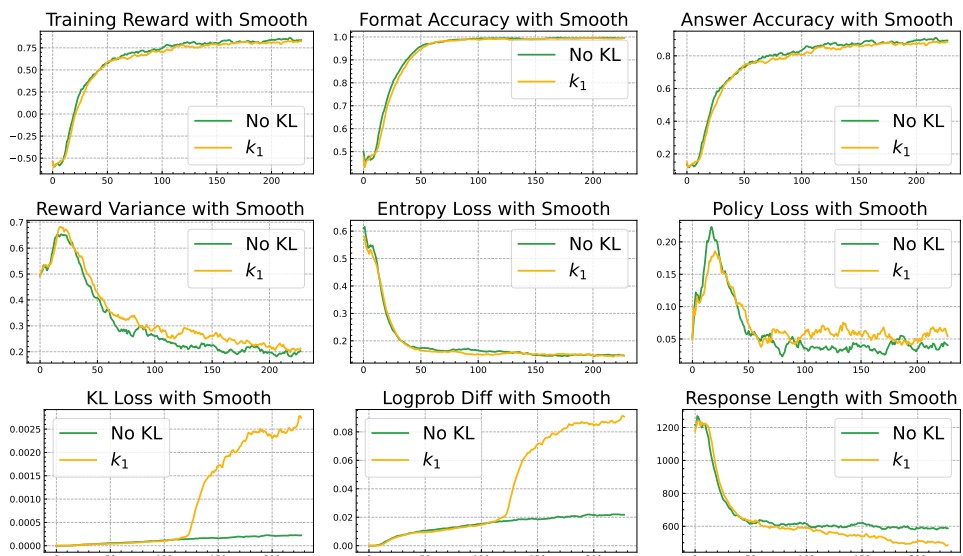

Figure 4: Comparison of "$k_1$ as loss'' versus no KL regularization. The training dynamics are nearly indistinguishable, empirically confirming the theoretical prediction from Section 5: '$k_1$ as loss' is ineffective as a KL regularizer due to its gradient's independence from the reference model and its zero-mean gradient expectation.

The empirical results on the larger 7B-scale model in Figure 4 further corroborate our theoretical analysis of '$k_1$ as loss'. As derived in Section 5, the gradient of this loss term is fundamentally unsuitable for regularization: it is independent of the reference policy $\pi_{\text{ref}}$ and has an expectation of exactly zero under on-policy sampling. In practice, this is equivalent to injecting a scaled score function, $\beta \cdot \nabla_\theta \log \pi_\theta$, which introduces zero mean but potentially high variance noise. Although the expected update direction remains unchanged, increased gradient variance can occasionally produce large deviations in a single update step. This phenomenon is clearly reflected in the experimental curves. As shown in the **KL Loss** and **Logprob Diff**, the actor model under '$k_1$ as loss' not only fails to stay closer to the reference model but, in fact, drifts further away at later stages. The trajectories of both settings are initially aligned, but the '$k_1$ as loss' variant suddenly diverges, indicating a sharp fluctuation induced by variance. Although the final task-level performance (e.g., reward/accuracy) remains broadly similar, achieving such results requires substantially higher KL magnitudes, which is ultimately inefficient and provides no meaningful regularization. In short, applying '$k_1$ as a loss' on a larger model scale is not only ineffective, but also counterproductive, as it destabilizes training and weakens alignment with the reference model.

The empirical results on the 7B scale model in Figure 5 highlight the contrasting behaviors of '$k_2$ as loss' and '$k_3$ as loss'. Under the same coefficient, '$k_2$ as loss' imposes a visibly stronger constraint: both the KL Loss and Logprob Diff curves remain consistently lower than those of '$k_3$ as loss', indicating that the actor stays closer to the reference model. In contrast, '$k_3$ as loss' tends to

diverge more during training, as seen from higher KL magnitudes and larger log-probability differences. This divergence is further reflected in the response length, where $k_3$ produces shorter and more variable outputs, suggesting weaker control over generation. Although '$k_3$ as loss' sometimes achieves higher Reward and Accuracy in the early and middle phases, this advantage comes at the cost of instability. The Reward Variance and Policy Loss under '$k_3$ as loss' are substantially higher, showing that its weaker constraint allows for larger fluctuations during optimization. In comparison, '$k_2$ as loss' provides a more stable training trajectory, maintaining a lower variance and keeping the policy tightly aligned with the reference. These results imply that, on larger model scales, '$k_2$ as loss' is the more effective choice for consistent and controlled regularization, while '$k_3$ as loss' risks greater drift and instability despite temporary performance gains.

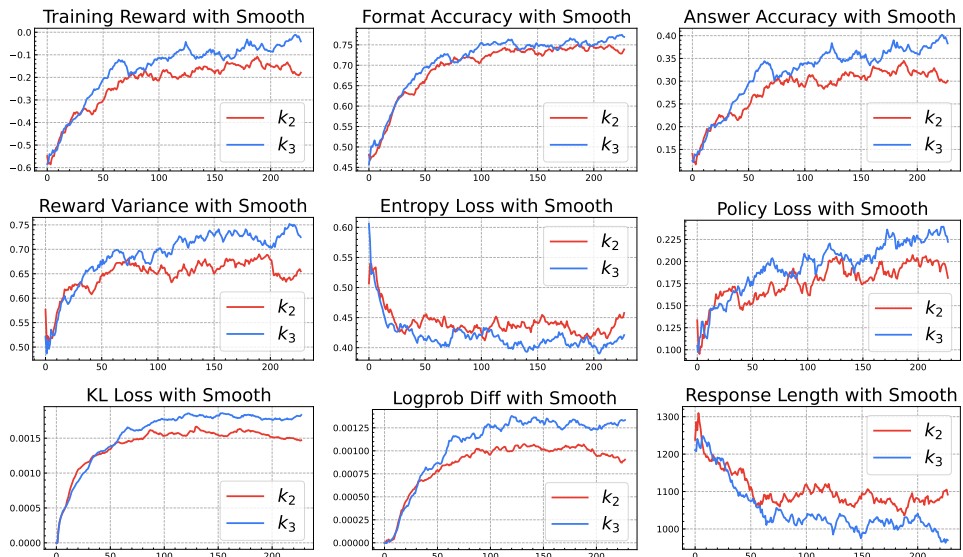

Figure 5: Comparison of the principled "$k_2$ **as loss**"' against its first-order surrogate "$k_3$ **as loss**"'. Both variants effectively constrain the policy, but '$k_2$ **as loss**' demonstrates superior regularization properties, maintaining a tighter coupling to the reference policy and yielding a more stable optimization path, evidenced by lower reward variance.

## M    DOWNSTREAM BENCHMARK PERFORMANCE

The main results on the math and general reasoning benchmarks are summarized in table 2. Several consistent patterns emerge across the 7B and 1.5B models.

First, the setting with '$k_1$ **as loss**' does not provide a significant benefit of regularization. This aligns with our theoretical analysis, which suggests that the expected gradient vanishes, thereby failing to constrain the actor with respect to the reference model. Empirically, its performance remains very close to the baseline 'RL without KL' in both mathematical and general domain tasks, demonstrating that '$k_1$ as loss' is ineffective as a regularizer.

Second, under the same coefficient, the behaviors of '$k_2$ as loss' and '$k_3$ as loss' diverge significantly. The '$k_2$ as loss enforces a much tighter constraint on the actor: the model remains closer to the reference policy, but this tighter coupling comes at the cost of substantially degraded performance, as reflected in both math reasoning (e.g., AIME (Li et al., 2024), AMC (Li et al., 2024), MATH-500 (Hendrycks et al., 2021)) and general reasoning benchmarks (e.g., ARC-c (Clark et al., 2018), GPQA* (Rein et al., 2024), MMLU-Pro (Wang et al., 2024)). In contrast, '$k_3$ as loss' imposes a weaker constraint and allows the model to drift more, as also observed in training dynamics (higher KL and log-prob differences). Although '$k_3$ as loss' appears slightly better than '$k_2$ as loss' in the final benchmark scores, its more divergent behavior highlights the lack of effective regularization and greater instability.

Taken together, these results confirm our theoretical expectations: '$k_1$ as loss' does not act as a constraint; '$k_2$ as loss' imposes stronger and rigid regularization that suppresses overall performance; and '$k_3$ as loss', though less restrictive, allows excessive divergence, may leads to unstable training.

Table 2: Main experiment results on math and general reasoning benchmarks based on **Qwen2.5-Math-7B** and **Qwen2.5-Math-1.5B**.

| Model | Math Reasoning Performance | | | | | | General Domain Reasoning Performance | | | |
|---|---|---|---|---|---|---|---|---|---|---|
| | AIME 24/25 | AMC | MATH-500 | Minerva | Olympiad | Avg. | ARC-c | GPQA* | MMLU-Pro | Avg. |
| Qwen2.5-Math-7B | 11.5/4.9 | 31.3 | 43.6 | 7.4 | 15.6 | 19.0 | 18.2 | 11.1 | 16.9 | 15.4 |
| RL w/o KL | 20.5/14.4 | 55.6 | 78.6 | 36.8 | 42.4 | 41.4 | 81.7 | 33.8 | 46.9 | 54.1 |
| RL w/. $k1$ as loss | 19.1/11.6 | 56.0 | 80.6 | 40.8 | 43.0 | 41.8 | 79.7 | 29.8 | 45.1 | 51.5 |
| RL w/. $k2$ as loss | 15.4/7.5 | 48.5 | 64.2 | 16.9 | 24.9 | 29.6 | 31.3 | 15.2 | 27.1 | 24.5 |
| RL w/. $k3$ as loss | 19.0/7.3 | 48.9 | 65.4 | 18.8 | 29.0 | 31.4 | 29.6 | 19.2 | 27.7 | 25.5 |
| Qwen2.5-Math-1.5B | 7.2/3.6 | 26.4 | 28.0 | 9.6 | 21.2 | 16.0 | 3.5 | 4.0 | 2.5 | 3.3 |
| RL w/o KL | 12.5/4.8 | 43.7 | 66.8 | 28.3 | 31.9 | 31.3 | 43.7 | 19.2 | 23.1 | 28.7 |
| RL w/. $k1$ as loss | 13.8/4.7 | 41.5 | 68.0 | 25.7 | 31.9 | 30.9 | 36.6 | 18.2 | 21.0 | 25.3 |
| RL w/. $k2$ as loss | 7.0/5.5 | 35.2 | 52.8 | 14.7 | 29.0 | 24.0 | 7.8 | 7.6 | 4.9 | 6.8 |
| RL w/. $k3$ as loss | 7.7/3.8 | 34.9 | 54.2 | 15.8 | 28.0 | 24.1 | 11.3 | 8.1 | 5.5 | 8.3 |

# N    STATEMENT ON THE USE OF LARGE LANGUAGE MODELS

We used LLMs solely for language polishing and editing. All retrieval of related work, algorithmic design, and theoretical derivations are carried out by the authors.

# O    IMPACT

Our "$k_2$ as loss" formulation has been merged into OpenRLHF and has been adopted and cited by Reinforce++. Prorl also integrates it with periodic resetting of the reference model. By providing a gradient-correct, off-policy–ready treatment of KL regularization, our work clarifies long-standing ambiguities and offers practical guidance for building stable, effective, and reproducible RLHF systems. We anticipate that these contributions will enable the community to design more reliable training pipelines and make significant advances in the field.

