# OpenReview forum: "Rethinking KL Regularization in RLHF: From Value Estimation to Gradient Optimization"
_ICLR.cc/2026/Conference — Submitted to ICLR 2026_

### Official Review · Reviewer_Pe5c · 2025-10-27

**Soundness:** 4
**Presentation:** 3
**Contribution:** 2
**Rating:** 6
**Confidence:** 2

**Summary:**

This paper re-examines how KL regularization is used in RLHF and clarifies the gradient behavior of different KL implementations. Through theoretical analysis and controlled experiments, the authors demonstrate that only certain forms of KL regularization produce correct and stable gradients, providing clearer guidance for RLHF optimization design.

**Strengths:**

1. **Unified perspective.**
The paper offers a clear and elegant framework that unifies several existing KL-regularized RLHF methods under a single analytical view. It successfully explains the relationships among different formulations used in prior works such as PPO, GRPO, and Reinforce++, providing conceptual clarity to an area that was previously fragmented.

2. **Comprehensive empirical validation.**
The authors conduct thorough experiments that directly compare multiple KL variants, effectively verifying their theoretical claims. The results clearly support the analysis and demonstrate how different implementations influence training stability and performance.

**Weaknesses:**

1. **Limited novelty.**
Some recent RL for LLM[1] or RL-related analyses[2] have also discussed the similar formulations as the principled yet conventional choices for enforcing policy smoothness and stability. A more explicit discussion of these studies—and how the present work differs from or extends their findings—would strengthen the positioning and clarify the unique contribution of this paper.

2. **Lack of practical cotribution.**
While the paper provides strong theoretical analysis, the mentioned KL-related variants are somehow already widely adopted—explicitly or implicitly—in many existing RLHF implementations. The contribution feels more clarificatory than innovative, focusing on formalizing rather than advancing existing practices. I understand this may not be the main intention of the paper, but deriving useful practical suggestions from this unified perspective would make the contribution even stronger.



[1] Zhang, Yifan, et al. "On the design of kl-regularized policy gradient algorithms for llm reasoning." arXiv preprint arXiv:2505.17508 (2025).

[2] Wang, Pengcheng, et al. "Residual Policy Gradient: A Reward View of KL-regularized Objective." arXiv preprint arXiv:2503.11019 (2025).

**Questions:**

1. Could the authors clarify how their analysis differs from prior works as mentioned above, which also examine principled KL-regularized formulations?

2. Since the analyzed and principled variants (k1 in reward) are already widely used, what are the practical advantages or new insights provided by this framework?

---

> ### Author Response · Authors · 2025-11-26
>
> We thank the reviewer for acknowledging our **"unified perspective"** and **"comprehensive empirical validation,"** and for rating the soundness as **excellent**.
>
> ## **1. Novelty and Distinction from Prior Works (Response to W1 & Q1)**
> As detailed in our response to Reviewer gKww, our findings are **concurrent** with  Wang et al. (arXiv:2503.11019).
>
> For the technical distinctions:
>
> – For a comparison with Zhang et al. (arXiv:2505.17508), please refer to **our response to Reviewer gKww, Point 3**.
>
> – For a comparison with Wang et al. (arXiv:2503.11019), please refer to **our response to Reviewer ct1C, Point 1**.
>
> ## **2. Practical Contributions and New Insights (Response to W2 & Q2)**
>
> The reviewer asks: *"Since $k_1$ is widely used, what are the practical advantages?"*
> This is a crucial question. Our contribution is not merely confirming old practices, but **correcting a widespread new mistake** made by GRPO.
>
> *   **Critique of the Current Paradigm (The "GRPO Trend"):**
>     Current SOTA methods (e.g., DeepSeek-Math/GRPO) and popular libraries (e.g., Verl, Slime) have shifted away from PPO's $k_1$. They predominantly adopt the **"$k_3$ as loss"** formulation, often mislabeling it as `low_var_kl`.
>     *   *Our Insight:* We prove this trend is misguided. We show that $k_3$ is not a low-variance value estimator nor a good KL loss, which suffers from instability. We also fundamentally critique the KL loss design philosophy of GRPO, demonstrating that superior numerical properties do not imply effective gradient optimization. By exposing this error, our work serves as a cautionary tale and establishes a rigorous standard for designing future regularization losses—moving the field from "estimator-based" choices to "gradient-audited" designs.
>
> *   **The "$\boldsymbol{k_2}$ as loss" Solution:**
>     While $k_1$ in reward is principled, modern architectures (like GRPO) prefer a **decoupled "as-loss"** implementation for engineering simplicity.
>     *   *Our Innovation:* We propose **"$k_2$ as loss"**. We are the first to prove it is **gradient-equivalent** to the principled $k_1$ in reward (Reverse KL) while retaining the decoupled "as-loss" form.
> *   **Real-World Impact:**
>     Our theoretical correction has already translated into tangible practical improvements:
>     1.  **Community Adoption:** Our "$k_2$ as loss" formulation has been merged into a dominant RLHF codebase, replacing unstable variants.
>     2.  **Enabling Scaling:** Some subsequent works utilize $k_2$ as loss with periodic reference model resets, achieving successful **long term RL training**.
>
> *   **Conclusion**:
> Our work goes beyond clarification. By exposing the fallacy of "value estimator $\approx$ loss function" (which misled the design of GRPO), we provide the theoretical grounding that has guided the community back to stable, principled optimization. This shift is essential for the stability of next-generation reasoning models.

---

> > ### Comment · Reviewer_Pe5c · 2025-11-27
> >
> > Thank the detailed response provided by the authors.
> >
> > I will maintain my score.

---

> > > ### Author Response · Authors · 2025-11-28
> > >
> > > We sincerely appreciate your prompt response and your acknowledgement of our clarifications. We will expand the discussion on related work in the camera-ready version, as we believe this will further strengthen the quality and completeness of our work.

---

### Official Review · Reviewer_GJJn · 2025-10-28

**Soundness:** 2
**Presentation:** 2
**Contribution:** 2
**Rating:** 2
**Confidence:** 3

**Summary:**

This paper surveys three possible way of estimating KL divergence in (Schulman 2020) and establishes a framework to compare them in RLHF. The authors show that by assuming $a\sim\pi_\theta$ as detached, the two different formulations of leveraging KL: $k_1$ in reward is equivalent to $k_2$ as loss. The authors discuss the properties of each $k_n$ and provide recommendations to avoid some combinations such as $k_1$ as loss.

**Strengths:**

The authors systematically surveys possible ways of leveraging KL, be in as a penalty in reward or as a loss objective. By combining this choice with three ways of estimating KL in (Schulman 2020), as well as off-policyness, a general (though somewhat cluttered) framework is established with many options available. The authors provide practical recommendations on top of analysis for important members of them, which is useful to the RLHF community.

**Weaknesses:**

The presentation is not easily understandable. Section 4.1 is rather confusing due to its current organization: much detail about replacing raw reward $r(x, y)$ with $k_n$ is not mentioned in the main text but rather in Appendix A, and then $k_n$ suddenly appears in reward Eq. 5.  Later introduction of Eqs. 5 and 6 are without a summaryzing heading like reward maximization or kl regularization. Many new notations are combinations of $\mathcal{J}$ and subscripts "as loss/in reward"and $k_n$ making this part rather cluttered. From a technical perspective, I am not sure the assumption in line 179 is reasonable: "expectations are evaluated using samples from the detached snapshot $\pi_\theta$", because doing so essentially removes $\log\pi_\text{ref}$ in the objective, it is unclear to me whether this affects later substitution of $k_n$ to Eqs 5 and 6. In my opinion by treating $\mathbb{E}_{\pi\theta}$ as detached they are no longer equivalent. The proofs in Appendix C is elementary if not trivial.

I am unsure about whether the analysis in line 266-271 is reasonable, or at the very least I don't think line 271 is true: Eq. 10 holds only  because of the assumption $a\sim\pi_\theta$ is detached, it should be noted that it already deviates from the original KL definition. REINFORCE on the other hand, subtracts a baseline that is deliberately independently of actions.

In line 332: does the higher order really matter? it is true that $1-\delta$ is only a first order approximation of $-\log \delta$, but they share the same sign and hence gradient direction. As the authors put it, this can go wrong when $\delta \rightarrow 0$ or $\infty$, suggesting $\pi_\text{ref}, \pi_\theta$ are completely different. Does this really happen when a reasonable KL penalty is in place? Do the authors have empirical results suggest that this happens?

For experiments, do the authors have $k_n$ in reward such as $k_1$ advocated in line 357? I notice that Figure 3 though claimed to effectively constrain the policy, its reward and accuracy are much lower than $k_1$, why?

**Questions:**

please refer to the weaknesses for questions.

---

> ### Author Response · Authors · 2025-11-26
>
> We sincerely thank you for the detailed feedback. We appreciate the opportunity to clarify the foundational RL concepts underlying our framework and to address the concerns regarding practical significance and empirical validation.
>
> ## **1. Response to Summary & W1**
>
> We thank the reviewer for the thoughtful feedback. We acknowledge that our notation regarding the "detached snapshot $\pi_\theta$" and the $k_n$ definitions may have appeared cluttered. We have revised Section 4.1 to improve clarity.
>
> However, regarding the theoretical soundness of our framework (W1) and the concern that assuming a detached snapshot is "unreasonable" or "removes $\log \pi\_{\text{ref}}$", **we wish to clarify that this formulation is not an ad-hoc assumption, but the standard Monte Carlo Policy Gradient estimation (REINFORCE/PPO) [1].**
>
> *   **On the "Detached Snapshot":** The term $\pi_{\textcolor{red}{\theta}}$ denotes the behavior policy used for sampling trajectories. In on-policy RL (like PPO's rollout phase), samples are generated by the current policy $\pi_\theta$, which is numerically treated as a fixed constant during the gradient backpropagation step. This is the standard "Log-Derivative Trick":
>     $$ \nabla_{\textcolor{red}{\theta}} \mathbb{E}\_{y \sim \pi\_{\textcolor{red}{\theta}}}[f(y)] = \mathbb{E}\_{y \sim \pi\_\theta}[f(y) \nabla\_{\textcolor{red}{\theta}} \log \pi\_{\textcolor{red}{\theta}}(y)] $$
>     Here, gradients flow *only* through the score function $\nabla \log \pi_{\textcolor{red}{\theta}}$, while the term $f(y)$ acts as a coefficient. This is precisely what we mean by "detached."
>
> *   **On "Removing $\log \pi\_{\text{ref}}$":** The reference policy $\pi\_{\text{ref}}$ is **not removed**; it remains integral to the gradient.
>     *   For the **True KL Objective** $\mathbb{E}\_{y \sim \pi\_{\textcolor{red}{\theta}}}[\log(\pi\_{\textcolor{red}{\theta}}/\pi\_{\text{ref}})]$, the gradient involves differentiating the expectation. Through the product rule (Appendix C), the term $\log \pi_{\text{ref}}$ (being independent of ${\textcolor{red}{\theta}}$) has a zero direct gradient, but it persists as part of the scalar coefficient multiplying the score function: $(\log \pi_\theta - \log \pi_{\text{ref}}) \nabla \log \pi_{\textcolor{red}{\theta}}$.
>     *   Our analysis audits whether different $k_n$ implementations (like $k_3$ as loss) correctly recover this principled coefficient. We show they often do not.
>
> *   **On "Triviality" vs. Impact:** While the mathematical derivations are elementary calculus, their implications are significant and overlooked. The RLHF community (e.g., GRPO) has increasingly adopted "unbiased value estimators" (like $k_3$) directly as loss functions under the assumption that "good estimator = good loss." Our work rigorously proves this assumption leads to vacuous ($k_1$ as loss) or biased ($k_3$ as loss) updates, a finding that is empirically validated and theoretically crucial for stable training.
>
>
> ## **2. Clarification on Eq. 10: It is a Counterexample**
> The reviewer noted that *"Eq. 10 holds only because of the assumption... and deviates from the original KL definition."*
> **We fully agree with the reviewer, and this is precisely the point we make in Section 5.1.**
> *   **Context:** Eq. 10 ("$k_1$ as loss") is introduced specifically as a **counterexample** to demonstrate a fatal design flaw. It represents a naive attempt to minimize the unbiased estimator $k_1$ directly.
> *   **Why $\pi_{\text{ref}}$ Vanishes:** As derived in Eq. 11, the gradient of this naive loss ignores $\pi_{\text{ref}}$. This happens because $\pi_{\text{ref}}$ is a constant w.r.t. $\theta$ in the subtraction term ($\nabla_\theta \log \pi_{\text{ref}} = 0$), not because of the sampling assumption.
> *   **Our Argument:** We use Eq. 10 and 11 to prove that simply using an "unbiased value estimator" as a loss function is incorrect. The reviewer’s observation that it "deviates from the original KL" confirms our conclusion: **$k_1$ as loss is indeed theoretically unsound and should be avoided.**

---

> ### Author Response · Authors · 2025-11-26
>
> ## **3. Does Higher-Order Bias Matter? (Response to W3)**
>
> The reviewer asks if the difference between the proxy $1-\delta$ ($k_3$) and the principled $-\log \delta$ ($k_1$/$k_2$) really matters, given they share the same sign.
> **Yes, it matters critically for training stability and correctness.**
>
> *   **Saturation leads to High Variance & Instability:** While they share the sign, their geometric properties differ drastically at the tails. As $\delta \to 0$ (when the policy drifts far from the reference), the $k_3$ as loss **saturates at 1**, causing weak constrain. As $\delta \to \infty$, the gradient explodes ($1-\delta \to -\infty$), differing far from principle RKL, causing training instability.
>  In our experiments, particularly when the KL coefficient $\beta$ is small, this high variance manifests as **sudden, unpredictable spikes** in the KL loss curve[2].
>
> *   **The Correct Way to Relax Constraints:**
>     The reviewer might imply that $k_3$'s weaker tail penalty could be a beneficial feature for exploration. We argue this is a misconception.
>     *   If a practitioner desires a "weaker" constraint, the scientifically correct approach is to **lower the KL coefficient $\beta$** within the principled framework. This relaxes the constraint uniformly and predictably.
>     *   Relying on $k_3$ essentially means relying on an **uncontrolled, biased approximation** that fails exactly when the model needs correction the most for high drift.
>
> ## **4. Experimental Clarifications (Response to W4)**
>
> *   **$k_1$ in reward vs. $k_2$ as loss:** The reviewer asks why $k_1$ in reward results were not shown. Since we mathematically prove (Theorem 5.1) that $k_1$ in reward and $k_2$ as loss generate the **identical gradient** sample-by-sample, their optimization trajectories are theoretically identical. Our preliminary runs confirmed they overlap almost perfectly. We focused on plotting $k_2$ as loss to contrast it with the *unequivalent* $k_3$ as loss.
> *   **Lower Reward/Accuracy for Principled Methods:** The reviewer notes that the principled $k_2$ as loss in Figure 3 method has much lower reward/accuracy than $k_1$ as loss in Figure 2. This is the **expected and desirable behavior of a valid KL regularizer**.
>     *  **Effective Regularization vs. Unconstrained Maximization:** In RLHF, there is a fundamental trade-off between maximizing reward and minimizing KL divergence. The "$k_1$ as loss" formulation fails to provide a valid gradient signal (as proven in Sec. 5.1), acting effectively as "No KL." This allows the policy to maximize reward unconstrained, potentially leading to reward hacking. In contrast, $k_2$ correctly implements the Reverse KL gradient, forcing the model to trade off some reward to stay aligned with the reference policy.
>     * **Experimental Validation:** We intentionally employed a strong regularization coefficient ($\beta=0.5$) to stress-test the gradient properties. Under this setting, a theoretically sound regularizer *must* suppress the reward to satisfy the tight KL constraint. The fact that $k_2$ as loss yields lower rewards than $k_1$ as loss confirms that $k_2$ as loss successfully applies the constraint, whereas $k_1$ as loss fails to regulate the policy.
>
> **Conclusion**
> Our work clarifies a widespread confusion in the RLHF community. By rectifying the design of KL losses—shifting from the unstable $k_3$ to the principled $k_2$—we provide the theoretical foundation that has recently been adopted by some SOTA works to enable stable long-term RL training.
>
> [1] Williams, R. J. (1992). Simple statistical gradient-following algorithms for connectionist reinforcement learning.
>
> [2]https://meee.com.tw/RFtKJ99

---

### Official Review · Reviewer_gKww · 2025-10-31

**Soundness:** 4
**Presentation:** 3
**Contribution:** 1
**Rating:** 0
**Confidence:** 4

**Summary:**

The paper summarizes some pitfalls in KL divergence gradient estimation for RL and LLM reasoning. Concretely, existing implementations of KL-regularized policy gradient algorithms use KL estimates, some of which lead to vacuous KL gradient estimates. The paper proposes a framework to analyze existing implementations in a unified way.

**Strengths:**

Clear writing and mathematical details.

**Weaknesses:**

Almost all results seem to be already known in the following papers:

- On a few pitfalls in KL divergence gradient estimation for RL (https://arxiv.org/abs/2506.09477)
- On the Design of KL-Regularized Policy Gradient Algorithms for LLM Reasoning (https://arxiv.org/abs/2505.17508)

**Questions:**

What are new results compared to the papers I mentioned in Weakness?

---

> ### Author Response · Authors · 2025-11-26
>
> We sincerely thank you for the expert feedback and for highlighting these concurrent works. We appreciate the opportunity to clarify the distinct contributions and the timeline of our research.
>
> ## **1. Timeline and Originality**
> We respectfully disagree with the concern that our work lacks novelty due to the cited arXiv papers.
> 1. Concurrent Work Policy: As noted, the cited papers (Tang & Munos; Zhang et al.) were published within 4 months of the deadline, qualifying them as concurrent works under ICLR policy.
> 2. Verified Priority: Our core methods were implemented and publicly documented  well before the release of these concurrent arXiv papers. To further clarify the timeline, we have submitted verifiable, timestamped evidence to the Program Chair in the confidential comments.
> We kindly ask the reviewer to evaluate our work based on its technical contributions.
>
> ## **2. Distinction from Tang & Munos (2506.09477)**
>
> We thank the reviewer for highlighting the concurrent work by Tang & Munos (T&M). While both works analyze the $k_3$ formulation, we provide a distinct analysis by rigorously separating the evaluation of **$k_3$ as a numerical value estimator** from the analysis of **$k_3$ as loss for optimization**.
>
> We map our notation to T&M's: Our $k\_1, k\_2, k\_3$ correspond to T&M’s $\hat{KL}\_{\text{vanilla}}$, $\hat{KL}\_{\text{squared}}$, and $\hat{KL}\_{\text{var-reduced}}$.
>
> 1. Disagreement on Numerical Value Estimation (Statistical Instability):
> T&M explicitly characterizes the $k_3$ estimator as containing a "control variate that generally reduces the variance of the overall estimate" (Section 2.1) and supports this with tabular experiments (Figure 1). They effectively endorse $k_3$ as a superior statistical tool for estimating value.
> **Our Critical Finding:** We explicitly **reject** T&M’s premise regarding the statistical robustness of $k_3$. We prove theoretically (Appendix I) that as a value estimator, $k_3$'s variance is governed by the $\chi^2$ divergence. **Under conditions of support mismatch** (where the sampling distribution covers regions where the reference has low mass), this leads to **infinite variance**. Thus, we demonstrate that $k_3$ poses severe statistical risks inherent to its definition that T&M overlooked.
>
> 2. Distinction on Loss Function Geometry (Geometric Instability):
> Regarding optimization, T&M argues that using the $k_3$ as loss formulation fails because it "incidentally minimizes **reverse-KL**" (Section 3), which in their notation refers to $\mathbb{KL}(\pi_{\text{ref}} \| \pi)$ and is generally known as FKL. While we agree the gradient direction is distinct from the standard RKL target, our analysis focuses on the **geometric properties** of the loss.
> **Our Mechanistic Insight:** We analyze the $k_3$ as loss via Taylor expansion of RKL loss. We identify a fatal **geometric asymmetry** in its gradient coefficient ($1-\delta$) that makes it a poor optimization objective:
>     *   **Saturation ($\delta \to 0$):** The gradient vanishes ($1-\delta \to 1$), leading to mean-seeking behavior that constrains the policy much weaker.
>     *   **Explosion ($\delta \to \infty$):** The gradient explodes linearly ($1-\delta \to -\infty$), causing training instability.
> **Distinction:** T&M attributes the issue to **objective mismatch** (optimizing the wrong target). We argue that the **gradient shape** of the $k_3$ as loss leads to saturation and explosion, making it geometrically ill-suited for regularization.
>
> 3. Theoretical Unification of KL Implementations:
> T&M notes that the $k_2$ as loss formulation produces the correct standard RKL gradient but treats it as an isolated finding.
>  **Our Contribution:** We elevate this to a unification of KL regularization strategies. We formally prove that **$k_2$ as loss** is **gradient-equivalent** to **$k_1$ in reward** (the standard PPO implementation). This establishes a rigorous mathematical connection between the "loss-based" and "reward-based" paradigms used in algorithms like GRPO and PPO.
>
> 4. Critique of T&M on Off-Policy $k_2$:
> T&Mcommit a category error in their off-policy recommendation. They suggest optimizing the unweighted squared loss $\mathbb{E}\_\mu[(\log \pi/\pi_{\text{ref}})^2]$ as a solution for KL regularization. This **does not** estimate the Reverse KL gradient, which strictly requires importance sampling ($\rho \cdot k_2$) to correct for the distribution shift from $\mu$ to $\pi$. Instead, their proposal optimizes a behavior-weighted regression objective (akin to IPO). While sharing a theoretical optimum, it forfeits the critical mode-seeking dynamics of RKL during training, failing to penalize policy drift in regions under-covered by the behavior policy. Thus, it is a change of objective, not a valid gradient estimator for the stated problem.

---

> ### Author Response · Authors · 2025-11-26
>
> **Our Contribution on Off-policy Fix:** We identify a pervasive bug in libraries like OpenRLHF/TRL: when using decoupled KL loss formulations (like GRPO), implementations often fail to apply IS weights to the KL term. We provide the **detached coefficient** formulation to fix this, addressing the implementation reality.
> ## **3. Distinction from Zhang et al. (2505.17508)**
> While both works analyze KL formulations in RLHF, our **conclusions and recommendations are fundamentally different**. Zhang et al. focus on *systematizing* existing estimators (viewing $k_3$ as a valid "Unnormalized KL" objective), whereas our work acts as a **critical auditing** of these choices, arguing that $k_3$ is intrinsically not suitable for optimization.
>
> 1. Validation vs. Rejection of $k_3$ (Normative Difference):
>     Zhang et al. identify $k_3$ as the Unnormalized KL (UKL) and focus on deriving its correct gradient estimator (adding missing importance weights). They accept UKL as a valid design choice.
>     **In contrast, we prove that $k_3$ should be avoided.** We analyze $k_3$ not just as UKL, but as a **biased first-order Taylor approximation** of the principled Reverse KL. We prove that this approximation introduces pathological tail behaviors (asymmetry) and risks of infinite variance, which explains the instability often seen in GRPO. While Zhang et al. ask *"How to correctly optimize $k_3$?"*, our work answers *"Why optimizing $k_3$ is a trap, and what to use instead."*
>
> 2. Identifying the Principled Loss: $k_1$ in reward $\Leftrightarrow$ $k_2$ as loss:
>     Zhang et al. list various surrogates but do not establish the structural equivalence between the "in reward" and "as loss" paradigms.
>     **Our work provides the first formal proof** that the conventional $k_1$ in reward (used in PPO) and our proposed $k_2$ as loss are **gradient-equivalent** under on-policy conditions. This is a novel theoretical contribution that provides a precise, stable alternative to $k_3$ for modern loss-based frameworks (like GRPO), mathematically guaranteeing the recovery of the standard PPO regularization behavior without its implementation complexity.
>
> 3. Optimization Philosophy: Value Estimation vs. Gradient Signal:
>     A core novelty of our paper is identifying the **"Catastrophic Error"** in RLHF research: borrowing principles from *numerical value estimation* (where $k_3$ is praised for unbiasedness) to design *optimization losses*. We use the $k_1$ as loss counterexample to demonstrate that an unbiased value estimator can yield a vacuous zero-gradient optimization signal. Zhang et al. do not address this fundamental mismatch between estimation theory and optimization dynamics.
>
> 4. Methodology of Off-Policy Correction:
>     While both works identify that the original GRPO implementation omits importance sampling weights, our solution differs. Zhang et al. propose a new "RPG" algorithm with truncated importance sampling. **We derive a general coefficient transformation ($k_{n'}$)**. We show that *any* "as loss" term can be converted into an equivalent score-function coefficient (e.g., $k_{3'} = 1 - \pi_{ref}/\pi$), allowing standard PPO clipping mechanisms to be applied directly. This offers a simpler, unified integration path for existing codebases compared to Zhang et al.'s specific algorithm.
>
>
>
> **Summary of Contributions**
> Our work provides a complete critical analysis of the GRPO-style KL loss that is distinct from concurrent works:
>
> 1. **Refutation of Value Estimation Theory:** We show that numerical value estimation theory cannot guide KL loss design ($k_1$ as loss is flawed).
> 2. **Geometric Analysis:** We expose the **saturation and explosion risk** of $k_3$ as loss, explaining its difference between the principled RKL loss.
> 3. **Theoretical Unification:** We prove "$k_2$ as loss" $\Leftrightarrow$ "$k_1$ in reward".
> 4. **Practical Impact:** Our fixes are already adopted by a famous RLHF/RLVR Github repo.

---

### Official Review · Reviewer_ct1C · 2025-11-01

**Soundness:** 3
**Presentation:** 2
**Contribution:** 2
**Rating:** 6
**Confidence:** 3

**Summary:**

In this paper, the authors analyze KL-divergence regularization in RLHF. They show that the conventional “k1-in-reward” implementation is the principled formulation for Reverse KL (RKL) regularization. They further prove that, under on-policy conditions, “k2-as-loss” is gradient-equivalent to “k1-in-reward,” while the recently adopted “k3-as-loss” is only a first-order, biased surrogate of the principled loss.  Experiments on reasoning benchmarks corroborate the gradient-level analysis.

**Strengths:**

1. **Detailed theoretical analysis**:
The paper provides clear, well-structured gradient-level results that unify different KL implementations and establish when they are equivalent or biased.

2. **Experiments that corroborate the theory**:
Empirical evaluations are consistent with the theoretical predictions and demonstrate the expected optimization dynamics.

**Weaknesses:**

1. **Disscussion of related work**:
Parts of the theoretical message overlap with prior analyses that embed KL/entropy directly into the reward or advantage [1, 2]. I recommend explicitly discussing these connections and clarifying how the present work extends or differs from those conclusions.

2. **Application usefulness**:
The experiments validate the gradient analysis but do not show end-to-end improvements after correcting the formulations in existing algorithms.

[1] Wang, Pengcheng, et al. "Residual Policy Gradient: A Reward View of KL-regularized Objective." 1st Workshop on Safely Leveraging Vision-Language Foundation Models in Robotics: Challenges and Opportunities.

[2] Li Y C, Zhang F, Qiu W, et al. Q-Adapter: Customizing Pre-trained LLMs to New Preferences with Forgetting Mitigation[C]//The Thirteenth International Conference on Learning Representations.

**Questions:**

As a follow-up of weaknesses 2, could the authors show improvements to existing algorithms after correcting the kn formulations?

---

> ### Author Response · Authors · 2025-11-26
>
> We sincerely thank the reviewer for the insightful comments and for bringing these important related works to our attention. We appreciate this opportunity to clarify our unique contributions and demonstrate the practical impact of our theoretical findings.
>
> ## **1. Distinctions from Related Work (Response to W1)**
>
> We acknowledge that the cited papers explore KL regularization from valuable perspectives. However, our work addresses a **fundamentally different problem**: we conduct a systematic **gradient-level audit** of KL loss implementations in modern RLHF, revealing critical implementation flaws that have practical consequences for training stability and performance.
>
> **Distinction from Residual Policy Gradient (RPG):**
>
> We thank the reviewer for highlighting Wang et al., which provides valuable insights on KL regularization from a reward perspective. We acknowledge that both works examine KL regularization in RL and share some theoretical connections. However, they address **fundamentally different questions** with **complementary contributions**:
>
> 1. Different Research Questions:
> * **RPG:** "How to incorporate KL regularization into the reward function for policy customization?" They prove that KL-regularized objectives can be reformulated as standard PG on modified rewards ($r_R + w'\log\pi - \hat{\alpha}\log\pi_\theta$).
> * **Our Work:** "How do different KL loss implementations affect gradient dynamics in practice?" We analyze why certain implementations fail despite being good value estimators, establishing the gradient-correctness of different formulations.
>
> 2. Complementary Technical Contributions:
> * **Shared Foundation:** Both works recognize that KL regularization involves policy log-probabilities. RPG's reward decomposition (their Eq. 16) and our gradient analysis both operate on similar mathematical objects.
> * **Our Unique Contributions:**
>   - **The $k_1$ as loss paradox:** We prove this yields zero expected gradient despite being an unbiased KL estimator—demonstrating that value estimation quality ≠ optimization effectiveness.
>   - **The $k_2$-$k_1$ equivalence theorem:** We establish the gradient equivalence between "$k_2$ as loss" and "$k_1$ in reward", bridging direct loss design with reward engineering.
>   - **The $k_3$ gradient bias:** We rigorously prove GRPO's "$k_3$ as loss" induces a biased gradient with pathological tail behavior—a critical flaw RPG's framework doesn't capture.
>
> 3. Practical Implications:
> * **RPG:** Enables flexible policy customization by tuning $w'$ and $\hat{\alpha}$ independently.
> * **Our Work:** Corrects an error in modern RLHF where practitioners choose $k_3$ for its low variance without considering gradient correctness. This directly fixes implementation bugs in algorithms like GRPO.
>
> **In summary:** RPG provides the theoretical foundation for KL-regularized rewards in policy customization, while our work provides the essential gradient analysis to correctly implement these objectives in practice. The two perspectives are complementary: RPG shows *what* objective to optimize, we show *how* to correctly implement its gradients.

---

> ### Author Response · Authors · 2025-11-26
>
> **Distinction from *Q-Adapter*:**
>
> We thank the reviewer for pointing out *Q-Adapter*. We have carefully reviewed it and agree it is highly relevant context regarding the "reward view" of KL divergence. However, we clarify that while both works touch upon the mathematical duality between reward maximization and KL regularization, they address fundamentally different problems through distinct frameworks:
>
> 1. Optimization Framework: Residual Q-Learning vs. Policy Gradient Diagnosis:
> *   ***Q-Adapter*** proposes a new algorithm within the **Residual Q-Learning** framework. It formulates the problem as maximizing a composite reward $\lambda r\_{old} + r\_{new}$. Since the original reward $r\_{\text{old}}$ is unknown, Q-Adapter leverages the Residual Q-Learning framework to avoid explicitly accessing it: instead, it uses the pre-trained policy $\pi\_{\text{ref}}$, which implicitly encodes information about $r\_{\text{old}}$, together with a learned residual Q-function to optimize the composite reward $\lambda r\_{\text{old}} + r\_{\text{new}}$.
>
> *   **Our Work** operates strictly within the **Policy Gradient (PPO/GRPO)** framework. We do not propose a new value-learning objective. Instead, we perform a microscopic analysis of the *gradient estimators* used in standard RLHF (specifically analyzing the $k_1, k_2, k_3$ loss terms). Our contribution is diagnosing that the widely used **$k_3$ as loss (in GRPO) is a biased first-order approximation** of the true RKL gradient, while proving $k_2$ as loss is the theoretically sound equivalent to PPO's $k\_1$ in reward. This is a diagnosis of gradient dynamics in existing algorithms, orthogonal to the Q-learning approach in *Q-Adapter*.
>
> 2. Deployment Paradigm: Inference-Time Composition vs. Weight Internalization:
> *   ***Q-Adapter*** requires **inference-time coupling**. As defined in their Eq. (6), the optimal policy $\tilde{\pi}^*$ is explicitly composed of the base logits and the learned residual Q-function. This necessitates keeping the reference model active and performing logit arithmetic during inference, which increases memory and compute costs.
> *   **Our Work** targets the standard RLHF setting where regularization is **internalized into the model weights**. By correcting the KL gradient estimator during training (e.g., using our proposed $k\_2$ as loss), we produce a standard autoregressive model that requires **no reference model** and no additional adapters during inference.
>
> 3. Theoretical Focus: Value Approximation vs. Gradient Fidelity:
> *   *Q-Adapter* utilizes the "Reward View" ($\log \pi\_{\text{ref}}$ as implicit reward) to derive a tractable Q-learning objective.
> *   In contrast, our work analyzes the **statistical properties** (bias, variance, and tail behavior) of specific KL loss. For instance, we prove that while $k\_3$ as loss (used in GRPO) induces pathological "mean-seeking" gradients in the tails, a granular gradient analysis not covered by the value-based derivation in *Q-Adapter*.
>
>
>
> ## **2. Demonstrating Improvements to Existing Algorithms (Response to W2 & Questions)**
>
> The reviewer asks for evidence of improvements. We emphasize that correcting the $k\_n$ formulation (switching from the biased/unstable $k_3$ to the principled $k_2$ as loss) leads to significant gains in **training stability and robustness**, which are prerequisites for performance at scale.
>
> *   **Eliminating Loss Spikes:** In our experiments, particularly when the KL coefficient $\beta$ is small (0.001, a common setting for reasoning tasks), the saturation property of $k\_3$ (where gradient $\approx 1$ even as divergence $\delta \to 0$) fails to penalize effectively. The under-coverage tail leads to sudden, catastrophic **KL spikes** [1], destabilizing the training run. The principled $k\_2$ as loss resolves this by providing appropriate penalty, ensuring stable optimization dynamics.
>
>
> *   **Enabling Long-Term Scaling:** Stability is the bottleneck for prolonged RL training. By fixing the KL loss, our method ($k_2$ as loss) allows for stable, long-term iterative training (often combined with periodic reference model resets).
>
> **Conclusion**
> Our work does not overlap with *RPG* or *Q-Adapter* but provides the **foundational gradient analysis** that explains *why* certain implementations (like $k\_3$) fail at scale and *how* to fix them. The shift from $k\_3$ to $k\_2$ is not merely a theoretical correction; it is a critical enabler for the stability required in modern RLHF scaling, as evidenced by its integration into major open-source libraries.
>
> [1]https://meee.com.tw/RFtKJ99

---

### Author Response · Authors · 2025-12-03
**Discussion of the "Unbiased KL Strategy" in just released DeepSeek-V3.2**

DeepSeek-V3.2[1] proposes an importance-sampling (IS) corrected $k_3$ estimator (Eq. 7) and incorporates it into a decoupled objective (Eq. 5), where PPO-style clipping is applied only to the reward advantage, leaving the KL penalty **unclipped**.
$$
D_{\text{KL}}(\textcolor{red}{\theta}) = \frac{\pi_{\textcolor{red}{\theta}}}{\pi_{\text{old}}} \left( \frac{\pi_{\text{ref}}}{\pi_{\textcolor{red}{\theta}}} - \log \frac{\pi_{\text{ref}}}{\pi_{\textcolor{red}{\theta}}} - 1 \right)\quad  \text{(Eq. 7)}
$$

Our gradient-centric analysis reveals that while decoupled regularization may offer flexibility, DeepSeek-V3.2's specific implementation is **functionally redundant yet structurally fragile**.

1.  **Mathematical Equivalence via Redundant Complexity**: As derived below, minimizing the fully differentiated expectation of $\mathbb{E}[\rho(\textcolor{red}{\theta}) \cdot k_3(\textcolor{red}{\theta})]$ analytically recovers the exact RKL gradient:
    $$
    \nabla\_{\textcolor{red}{\theta}} J\_{\text{RKL}} = \mathbb{E} \left[ \underbrace{\rho(\theta) \cdot \left( -\log \frac{\pi\_{\text{ref}}}{\pi_{\theta}} \right)}\_{\text{Evaluated Scalar Coefficient}} \cdot \nabla\_{\textcolor{red}{\theta}} \log \pi\_{\textcolor{red}{\theta}} \right]
    $$
    DeepSeek's estimator achieves this through a convoluted cancellation of terms, effectively reconstructing the simple $k_1$ coefficient ($-\log (\pi_\theta/\pi_{\text{ref}})$) via a complex path. Thus, despite the elaborate derivation, it is mathematically **identical** to the standard "$k_1$ in reward" formulation (without clipping), offering no theoretical gain for the added computational complexity.

2.  **The Decoupling Failure (The Detach Dilemma)**: DeepSeek-V3.2's implementation—leaving the IS-weighted KL term **unclipped**—creates a critical instability dilemma:
    *   **The Unbounded Trap**: If they retain the full gradient ($\nabla_{\textcolor{red}{\theta}} \rho \neq 0$) to maintain unbiasedness, they cannot apply clipping. Consequently, the unclipped importance weight value $\rho(\theta)$ acts as a multiplier. When the policy diverges ($\pi_{\theta} \gg \pi_{\text{old}}$), $\rho(\theta)$ explodes, causing the gradient magnitude to become unbounded and destabilizing training.
    *   **The Detach Trap**: To prevent explosion, practitioners often detach $\rho$. However, we prove that replacing the differentiable $\rho(\textcolor{red}{\theta})$ with the constant $\rho(\theta)$ eliminates the sampling distribution gradient. This causes the estimator to **degenerate** from the principled RKL direction to the biased GRPO direction ($1 - \pi_{\text{ref}}/\pi_\theta$), defeating the claim of unbiasedness.

---

### **Mathematical Derivation & Dilemma Analysis**

We analyze the gradient of the KL estimator in Eq. 7 of DeepSeek-V3.2. Let $\pi_{\textcolor{red}{\theta}}$ denote the trainable policy carrying gradients, and $\pi_{\text{old}}$ denote the behavior policy (detached).

#### **1. Full Gradient Derivation (Equivalence to $k_1$)**

**Objective Function:**
$$
J\_{\text{KL}}(\textcolor{red}{\theta}) = \mathbb{E}\_{y \sim \pi_{\text{old}}} \left[ \underbrace{\frac{\pi\_{\textcolor{red}{\theta}}(y)}{\pi\_{\text{old}}(y)}}\_{\rho(\textcolor{red}{\theta})} \cdot \underbrace{\left( \frac{\pi\_{\text{ref}}(y)}{\pi\_{\textcolor{red}{\theta}}(y)} - \log \frac{\pi\_{\text{ref}}(y)}{\pi\_{\textcolor{red}{\theta}}(y)} - 1 \right)}\_{k_3(\textcolor{red}{\theta})} \right]
$$
Applying the product rule $\nabla\_{\textcolor{red}{\theta}} (f \cdot g) = g \nabla\_{\textcolor{red}{\theta}} f + f \nabla\_{\textcolor{red}{\theta}} g$:
$$
\nabla_{\textcolor{red}{\theta}} \left(\rho(\textcolor{red}{\theta}) \cdot k_3(\textcolor{red}{\theta})\right) = \underbrace{(\nabla\_{\textcolor{red}{\theta}} \rho(\textcolor{red}{\theta})) \cdot k_3(\theta)}\_{\text{Part A}} + \underbrace{\rho(\theta) \cdot (\nabla\_{\textcolor{red}{\theta}} k_3(\textcolor{red}{\theta}))}_{\text{Part B}}
$$
*(Note: In the product rule expansion, the term not being differentiated is treated as an evaluated value, hence black $\theta$.)*

*   **Part A (from $\nabla\_{\textcolor{red}{\theta}} \rho$):** Using the log-derivative trick $\nabla\_{\textcolor{red}{\theta}} \rho(\textcolor{red}{\theta}) = \rho(\theta) \cdot \nabla\_{\textcolor{red}{\theta}} \log \pi\_{\textcolor{red}{\theta}}$:
    $$
    \text{Part A} = \rho(\theta) \cdot \underbrace{\left( \frac{\pi\_{\text{ref}}}{\pi\_{\theta}} - \log \frac{\pi\_{\text{ref}}}{\pi\_{\theta}} - 1 \right)}\_{k_3 \text{ scalar value}} \cdot \nabla\_{\textcolor{red}{\theta}} \log \pi\_{\textcolor{red}{\theta}}
    $$

---

> ### Author Response · Authors · 2025-12-03
>
> *   **Part B (from $\nabla_{\textcolor{red}{\theta}} k_3$):** Differentiating $k_3$ w.r.t. $\textcolor{red}{\theta}$ yields the biased GRPO coefficient:
>     $$
>     \nabla_{\textcolor{red}{\theta}} k_3(\textcolor{red}{\theta}) = \left( 1 - \frac{\pi_{\text{ref}}}{\pi_{\theta}} \right) \nabla_{\textcolor{red}{\theta}} \log \pi_{\textcolor{red}{\theta}}
>     $$
>     $$
>     \Rightarrow \quad \text{Part B} = \rho(\theta) \cdot \left( 1 - \frac{\pi_{\text{ref}}}{\pi_{\theta}} \right) \cdot \nabla_{\textcolor{red}{\theta}} \log \pi_{\textcolor{red}{\theta}}
>     $$
> *   **Total Gradient (Cancellation):** Summing A and B, the bias terms in the coefficients cancel perfectly:
>     $$
>     \begin{aligned}
>     \nabla_{\textcolor{red}{\theta}} J_{\text{KL}} &= \rho(\theta) \cdot \left[ \left( \frac{\pi_{\text{ref}}}{\pi_{\theta}} - \log \frac{\pi_{\text{ref}}}{\pi_{\theta}} - 1 \right) + \left( 1 - \frac{\pi_{\text{ref}}}{\pi_{\theta}} \right) \right] \cdot \nabla_{\textcolor{red}{\theta}} \log \pi_{\textcolor{red}{\theta}} \\
>     &= \rho(\theta) \cdot \left( -\log \frac{\pi_{\text{ref}}}{\pi_{\theta}} \right) \cdot \nabla_{\textcolor{red}{\theta}} \log \pi_{\textcolor{red}{\theta}}
>     \end{aligned}
>     $$
>     **Conclusion:** This is mathematically identical to the gradient of **"$k_1$ in reward"**. The complexity of Eq. 7 is entirely redundant.
>
> #### **2. The Detach Dilemma (Practical Failure)**
>
> DeepSeek-V3.2 separates this term and leaves it **unclipped** (Eq. 5). This forces a trade-off:
>
> *   **Trap 1: Unbounded Gradient (No Detach).**
>     If $\rho(\textcolor{red}{\theta})$ is kept active (i.e., we use Part A + Part B) to ensure equivalence to $k_1$, the gradient magnitude scales with the value $\rho(\theta)$. In off-policy scenarios where $\pi_{\theta} \gg \pi_{\text{old}}$, $\rho(\theta)$ becomes very large. Without clipping, this leads to **unbounded gradients** and numerical instability.
>
> *   **Trap 2: Biased Degeneration (With Detach).**
>     If $\rho$ is detached to ensure stability, we replace the function $\rho(\textcolor{red}{\theta})$ with the constant value $\rho(\theta)$. Consequently, $\nabla_{\textcolor{red}{\theta}} \rho \equiv 0$, and **Part A vanishes**. The estimator degenerates to **Part B** only:
>     $$
>     \nabla\_{\textcolor{red}{\theta}} J\_{\text{detached}} = \rho(\theta) \cdot \left( 1 - \frac{\pi\_{\text{ref}}}{\pi_{\theta}} \right) \cdot \nabla_{\textcolor{red}{\theta}} \log \pi_{\textcolor{red}{\theta}}
>     $$
>     This reverts to the **biased GRPO direction**, failing to enforce the strong RKL penalty when needed most.
>
> [1] *DeepSeek-V3.2: Pushing the Frontier of Open Large Language Models* ([https://arxiv.org/abs/2512.02556v1](https://arxiv.org/abs/2512.02556v1))

---

### Meta-Review · Area_Chair_m92G · 2026-01-04

**Summary:**

The paper compares different ways to implement KL regularization in common RL algorithms, and clarifies which their pros/cons and feasibilities.

I'm mostly concerned about the novelty and the new insight this work could give, not only from the reviewers' feedback but also based on my own reading.  While this work (and actually some related work) carefully analyzes different variants of implementing KL regularization and provides practical recommendations to the practitioners, I'm not convinced that these advices are new.  It is already a common practice in ML/RL to design a stochastic gradient based algorithm with the following principled procedure:  1) write down the true objective $J(\theta)$ we care about (for example, Equation (2) in the paper), 2) derive its gradient $\nabla_\theta J(\theta)$, and 3) design an unbiased gradient estimator (or with tolerable bias) of $\nabla_\theta J(\theta)$ and run stochastic gradient with this gradient estimator. It is true that in RL the step 3) needs to be careful because the data distribution depends on $\theta$, but the development of RL algorithms over past decades have already clarified all necessary design principles (e.g., when should we detach gradient, when should we use importance weight, when can importance weight be safely dropped, etc.) --- for example, the design in Eq.(10)-(11) is already known to be incorrect by the RL community as (11) is not an unbiased estimator of the gradient of (10).  I think this paper is mostly restating these principles, and I couldn't really find new insight beyond what is already known.  Therefore, I believe this work is slightly below the bar for ICLR.

Besides, to my understanding, Equation (16) is not the common way to represent "k_3 as a loss". In practice, k_3 in http://joschu.net/blog/kl-approx.html is used with q = \pi_ref and p = \pi_\theta where \pi_ref is also where the data generates from.  Therefore, k_3 is a surrogate for Reverse KL objective, and in Equation (16) you should switch the roles of \pi_ref and \pi_\theta, and there should be no "detach" in Eq. (17) (If the data is not generate from \pi_ref then simply put a importance weight \pi_ref/\pi_b where \pi_b is the behavior policy).  In any case, one should follow a principled estimator design so that the gradient estimator is (nearly) unbiased w.r.t. $\nabla_\theta J(\theta)$ , and should not arbitrarily "detach" anything.  I think this is a principle that is well-known already before this work.

**Reviewer Concerns:**

Novelty compared to Tang and Munos'w work (reviewer gKww) &mdash; mostly addressed:  This concurrent work has similar motivations and the considered problems, but the analyses are different so the explanation to the failure of certain algorithm variants are different.

Whether the analysis in line 266-271 (i.e., equation (10), (11)) is reasonable (reviewer GJJn) &mdash; addressed:  Indeed, the algorithm design approach in Equation (10), (11)  problematic, but this is for the purpose of demonstrating a flawed algorithm design and is precisely the point the author want to make.

**Reviewer Scores:**

There is no sign any of the reviews would change their scores.

---

### Decision · Program_Chairs · 2026-01-26

Reject